# Hazard assessment and hydrodynamic, morphodynamic, and hydrological response to Hurricanes Gamma and Delta on the northern Yucatan Peninsula

Alec Torres-Freyermuth[1,3], Gabriela Medellín[1,3], Jorge A. Kurczyn[1,3], Roger Pacheco-Castro[2,3], Jaime Arriaga[2,4], Christian M. Appendini[1,3], María Eugenia Allende-Arandía[1,3], Juan A. Gómez[1,3], Gemma L. Franklin[2,3], Jorge Zavala-Hidalgo[5]

[1]Laboratorio de Ingeniería y Procesos Costeros, Instituto de Ingeniería, Universidad Nacional Autónoma de México; Sisal, Yucatán, 97835, México.

[2]CONACYT- Laboratorio de Ingeniería y Procesos Costeros, Instituto de Ingeniería, Universidad Nacional Autónoma de México; Sisal, Yucatán, 97835, México.

[3]Laboratorio Nacional de Resiliencia Costera, Laboratorios Nacionales CONACYT, México.

[4]Faculty of Civil Engineering and Geosciences, Delft University of Technology, 2628 CN Delft, The Netherlands

[5]Instituto de Ciencias de la Atmósfera y Cambio Climático, Universidad Nacional Autónoma de México, Coyoacán, Ciudad de México, 04510, México.

*Correspondence to*: Alec Torres-Freyermuth (atorresf@iingen.unam.mx)

**Abstract.** Barrier islands in tropical regions are prone to coastal flooding and erosion during hurricane events. The Yucatan coast, characterized by karstic geology and the presence of barrier islands, was impacted by Hurricanes Gamma and Delta in

October 2020. Inner shelf, coastal, and inland observations were acquired simultaneously near a coastal community (Sisal, Yucatan) located within 150 km of the hurricanes' tracks. In the study area, Gamma moved slowly and induced heavy rain, mixing in the shelf sea, and strong winds ($>20$ m s$^{-1}$). Similar wind and wave conditions were observed during the passage of Hurricane Delta; however, a higher storm surge was measured due to wind setup and the drop ($<1000$ mbar) in atmospheric pressure. Beach morphology changes, based on GPS measurements conducted before and after the passage of the storms, show

alongshore gradients ascribed to the presence of coastal structures and macrophyte wracks on the beach face. Urban flooding occurred mainly on the back-barrier associated with heavy inland rain and the coastal aquifer's confinement, preventing rapid infiltration. Two different modeling systems, aimed at providing coastal flooding early warning and coastal hazards assessment, presented difficulties in forecasting the coastal hydrodynamic response during these seaward-traveling events, regardless of the grid resolution, which might be ascribed to a lack of terrestrial processes and uncertainties in the bathymetry

and boundary conditions. Compound flooding plays an important role in this region and must be incorporated in future modeling efforts.

# 1 Introduction

Barrier islands are highly vulnerable to terrestrial, atmospheric, and oceanic drivers associated with hurricane events and global climate change effects (Irish et al., 2010; Zinnert et al., 2017). The mean sea level rise rate is accelerating, and the proportion of major tropical cyclones has increased over recent decades due to the effects of climate change (Knutson et al., 2020). Therefore, knowledge of coastal dynamics during hurricane events and their impact on barrier islands is important to understand such coastal ecosystems natural resistance and resilience.

Significant advances have been achieved in recent decades regarding hurricane research. While Emanuel (2021) found an increased frequency of tropical cyclones in the North Atlantic, there is no clear trend in the increase of tropical cyclone frequency due to climate change. Nevertheless, most studies find an increased proportion of the most extreme events (i.e., categories 4 and 5 on the Saffir Simpson scale) in the context of climate change (Knutson et al., 2020), increasing the associated hazards by the second half of the century. Furthermore, a poleward migration of the location of the maximum lifetime intensity of tropical cyclones has been found (Kossin, 2014), as well as an increase in rapid intensification (Bhatia et al., 2019; Emanuel, 2017), increasing the hazards from tropical cyclones in higher latitudes and hence representing a challenge for emergency management.

Field observations provide valuable information to improve our understanding of the drivers and coastal response during extreme events (e.g., Valle-Levinson et al., 2002; Du et al., 2019; Mieras et al., 2021). Previous studies have demonstrated the important role of compound flooding driven by rain, storm surge, and groundwater processes (e.g., Wahl et al., 2015; Valle-Levinson et al., 2020; Housego et al., 2021). These studies are required to improve coastal hazard modeling and implement mitigation measures at both local and regional scales. However, concurrent measurements of meteorological, oceanic, coastal, and hydrological processes during extreme events are scarce in tropical systems; hence further research is warranted. Moreover, recent studies have pointed out the need to investigate storm impacts from an interdisciplinary point of view (Camelo and Tamayo, 2021). Considering the importance of numerical modelling for developing early warning systems and emergency management plans, an improvement in the understanding of the feedback between terrestrial-atmospheric-oceanic processes needs to be incorporated into numerical models to estimate their impacts in coastal areas. This is a challenging task for a low-lying (karstic) coast where the groundwater aquifer discharges towards the shore.

The Yucatan Peninsula is located in an area of high tropical cyclone activity. However, due to its geographic location, the northern coast (facing the Gulf of Mexico) is less prone to direct hurricane landfalls than other regions in the Gulf of Mexico and the western Caribbean Sea. For instance, of the 163 tropical storm events from 1842 to 2020, 64% landed on the eastern coast facing the Caribbean Sea (Rivera-Monroy et al., 2020), compared to 35% on the northern coast. Human settlements on a barrier island along the northern Yucatan coast are mainly devoted to artisanal fisheries (Paré and Fraga, 1994) and

recreational beach houses (Meyer-Arendt, 2001). These coastal communities are highly vulnerable to storm events due to their high exposure and potential communication breakdown with the inland due to high water levels.

Concurrent observations of atmospheric, terrestrial, and oceanic effects and coastal impacts during tropical cyclones are rare or inexistent in the region. Thus, this study investigates the effects and impacts of hurricane passages, specifically the tropical cyclones Delta and Gamma, on a barrier island located on the northwestern Yucatan coast for the first time. The observations of this coastal community are relevant as they represent the environmental, morphological, anthropogenic, and ecological conditions prevailing in this region (i.e., northern Yucatan Peninsula).

The outline of this paper is the following. An overview of the study area and the meteorological events (Gamma and Delta) is provided in Section 2. Section 3 describes the materials and methods employed in this work, including the data acquisition and analysis, and the implementation of the numerical models. Section 4 presents the observations of atmospheric, oceanic, and hydrological conditions associated with tropical storms and their impact on the coast of Sisal (Yucatan). Moreover, the capabilities and limitations of hydrodynamic numerical models for simulating waves and extreme water levels are investigated.

Finally, discussions (Section 5) and concluding remarks (Section 6) are presented.

**2 Study area**

The study area is the town of Sisal, Yucatan, located on the southeastern Gulf of Mexico coast (N 21° 09' 56.20'', W 90° 02' 26.44''). Sisal is a small community situated on a barrier island on the northwestern Yucatan Peninsula (Figure 1a), 50 km

from the city of Merida, with a population of less than 2,000, dedicated mainly to fishing activities and, more recently, to eco-tourism. The main infrastructure found in this community is a sheltered port devoted to artisanal fisheries and a local road passing through the wetland, connecting the barrier island with the hinterland. The area is of high ecological importance due to its biological diversity and because it is surrounded by two natural parks, *Ciénegas y Manglares de la Costa Norte de Yucatán* and the State Reserve of *El Palmar* to the west. Emblematic species such as jaguars (*Panthera onca*), crocodiles

(*Crocodylus moreleti*), flamingos (*Phoenicopterus ruber*), and sea turtles (*Eretmochelys imbricate* and *Chelonia mydas*), among others, are found in the tropical dry forest, wetlands, and beaches surrounding this coastal town.

The study area is characterized by intense sea breeze events (Figueroa-Espinoza et al., 2014) and winter storms (fall-winter) associated with Central American Cold Surge Events (Medina-Gómez & Herrera-Silveira, 2009; Kurzcyn et al., 2021). Typical winter storm wave conditions reach $H_s > 2$ m and $T_p > 8$ s from the NNW. Sea breezes drive low-energy high-angle short-period

waves ($H_s < 1$ m, $T_p = 3.5$ s, NE) that drive a persistent westward alongshore current (Torres-Freyermuth et al., 2017). Across the Yucatan shelf, winds and mesoscale circulation drive the currents mainly toward the west (Enriquez et al., 2010; Torres-Freyermuth et al., 2017). The mean sea level in this area presents a seasonal variability ascribed to alongshelf currents on the western Gulf of Mexico and low-frequency atmospheric pressure variability (Zavala-Hidalgo et al., 2003). The tidal regime in the study area is diurnal micro-tidal, with a spring tidal range of 0.75 m (Valle-Levinson et al., 2011).

The 2020 hurricane season was the most active season on record in the North Atlantic basin (Blunden and Boyer, 2021). Hurricane Gamma formed on October 2 in the western Caribbean Sea as a tropical depression southeast of Cozumel, Mexico. The hurricane made landfall on the eastern Yucatan Peninsula coast near Tulum, Mexico, on October 3 and then weakened into a tropical storm while crossing the Peninsula and reaching the Gulf of Mexico via the northern coast (Figure 1). Gamma interacted with the circulation associated with the formation of Hurricane Delta and moved southwestward to make landfall near Nichili, Mexico, and dissipated on October 6 (Latto, 2021). Hurricane Delta formed from a tropical wave in the Atlantic and attained the category of major hurricane on October 6, before undergoing rapid intensification and weakening before landing. It made landfall on October 7, on the northeastern portion of the Yucatan Peninsula near Puerto Morelos, Mexico, around 1030 UTC (20.848ºN, 86.875ºW). It continued its path inland and moved to the southern Gulf of Mexico by 1800 UTC 7 October with winds of 38 m/s, reaching the coast of Dzilam de Bravo (Yucatan) (21.393ºN, 88.892ºW) as a category 1 hurricane (160 Km/h). During its pass across the northern Yucatan Peninsula, it dumped 50-100 mm of rain. Delta made landfall near Creole, Louisiana, on October 9 as a category 2 hurricane and weakened to become an extratropical cyclone (Cangialosi and Berg, 2021).

## 3 Materials and methods

### 3.1 Data acquisition

Monitoring systems installed in the study area were employed to characterize the atmosphere, the ocean, beach morphology, and coastal aquifer response to the storm forcings (Figure 1b and 1c, and Table 1) west of the hurricane tracks.

A meteorological station, located at 10 m height and 100 m from the shoreline, measured the wind intensity and direction, the air temperature, and the atmospheric pressure at 10 Hz. A tidal gauge inside the port of Sisal recorded the mean sea level every 1 minute with an ultrasonic sensor. An Acoustic Doppler Current Profiler (ADCP), deployed 10 km offshore at 11 m depth, measured the wave parameters ($H_s$: significant wave height; $T_p$ = peak wave period; $\theta$: mean wave angle), the current profile, the sea surface height ($\eta$), and the sea bottom temperature ($T_b$). Waves were measured by taking 2048 samples at 2 Hz every hour, while the rest of the ADCP data were obtained every 20 min, averaging the first 60 s of the observations.

A beach monitoring program has been conducted regularly to investigate beach morphodynamics in the study area (Medellín and Torres-Freyermuth, 2019, 2021; Franklin et al., 2021). For this study, pre- (09/30/2020) and post- (10/14/2020) storm beach surveys were conducted along 40 equally-spaced cross-shore transects located east (updrift, P01-P20) and west (downdrift, P21-P40) of the Sisal port (see Figure 1c), encompassing a 4-km stretch of coast. Differential Global Positioning Systems (DGPS) with Real-Time Kinematics (RTK) were employed to conduct the beach survey. A reference station is located at a fixed location (top of a building), and the rover is carried on a backpack. Ground control points were measured at the beginning and end of each survey to correct the rover height. The DGPS-RTK measurements have a horizontal and vertical

accuracy of 0.010 and 0.020 m, respectively, as reported by the manufacturer. Additionally, a test was carried out to assess the method`s accuracy by comparing elevation measurements taken along a straight line on a parking lot with the GPS rover fixed on a pole with a bubble level against carrying the GPS rover in a backpack by two different users. Maximum differences in the vertical measurements were less than 0.04 m. Therefore, we expect that the maximum error associated with estimating the beach profiles' elevation is of that magnitude. The beach profiles started behind the foredune and reached a water depth of approximately 0.5-1.5 m depending on existing wave conditions, tidal level, and the presence of macrophytes.

A video monitoring system, placed at a height of 43-m, located 300-m west of the jetty and 100-m inland, acquired time exposure images over 10 minutes at 7.5~Hz of 2-km along the coast every 30 minutes during daytime hours (Arriaga et al., 2022). A gap in the images occurred between October 6 to 19 due to power failure. The spatial resolution of the different components of this camera system has been previously described by Arriaga et al. (2022). Given that here we are interested mainly in coverage, this result has to be translated into areas. For example, along the shoreline near the cameras (300 m easting), a pixel translates to an area of 0.01 $m^2$, whereas 1400 m easting a pixel covers an approximate size of 2.5 $m^2$. Finally, at the farthest point, near the pier, the resolution is in the order of 10 $m^2$. The calibration performed following Simarro et al. (2017), particularly the simplified mathematical model referred to as M2 in Simarro et al. (2020), resulted in an average reprojection error of 0.7 pixels.

The coastal aquifer response was characterized by groundwater pressure, temperature, and salinity measured every 30 minutes using pressure transducers (HOBO) installed in three monitoring wells (Figure 1b). The well W7a is located close to the Sisal port, whereas W5 and W4 are located 5 and 20 km inland, respectively. A detailed description of the monitoring wells can be found in Canul-Macario et al. (2020).

160    **Table 1.** Instruments and measured variables.

| Instrument | Measured variables | Location | Sampling interval |
| --- | --- | --- | --- |
| Meteorologial station | Air temperature, atmospheric pressure, wind intensity and direction, precipitation, and relative humidity. | Lat= 21.1645°N Lon=90.0484°W | 1 hour |
| Tide gauge | Sea level | Lat= 21.161°N Lon=90.048°W | 1 min |
| **Acoustic Doppler Current Profiler RDI** | Significant wave height, peak wave period, mean wave direction, and near-bed temperature. | Lat= 21.27529°N Lon=90.03711°W | 60 min |
| | Current profile ($u$ and $v$ components), $\eta$, $T_b$ | | 20 min |
| Video Camera System SIRENA | Timex images of a 2-km straight of coast | Lat= 21° 09 53N Lon=90. 02 48 W | 30 min |
| **Monitoring Wells (HOBOS pressure transducers)** | Water head and temperature of the coastal aquifer | W7a Lon=90.0468°W Lat=21.1630°N W5 Lon=89.9995°W Lat=21.1206°N W4 Lon=89.9689°W Lat=20.9748°N | 30 min |

## 3.2 Data analysis

165    Time series analyses, obtained from different *in situ* instruments, were carried out to characterize the main drivers and responses.

### 3.2.1 Monitoring wells

Pressure measured at the monitoring wells is converted to hydraulic head as follows: atmospheric pressure, $p_{atm}$, is subtracted from the pressure, $p$, measured at the well [kPa], and then converted to water column height $H$ [m], using a conversion factor of 0.102 using Eq. (1). Finally, the measurements are referenced to the same datum using the elevation of a known point at the well casing, $z_{well}$, and one measurement from this point to the water depth, $H_0$, at a known time $t_0$. The equations used are the following:

$$H(t) = 0.102\big(p(t) - p_{atm}(t)\big) \tag{1}$$

$$h(t) = H(t) - H(t_0) + (z_{well} - H_0) \tag{2}$$

where $h(t)$ is the hydraulic head [m] time series, that represents the water level referenced to a given datum for unconfined aquifers. To obtain the relative water head, the mean value is subtracted from the time series.

### 3.2.2 Acoustic Doppler Current Profiler

Bulk wave statistics ($H_s$, $T_p$, $\theta$) were computed from the 2048-s burst intervals using the *PUV* technique. For the ADCP measurements, the tidal signature was removed from the current profile and the sea surface height by applying a low-pass Lanczos filter, eliminating frequencies $\geq 1/48$ h. Ocean currents were then referenced to the angle of maximum variance, orienting them 25 degrees counterclockwise from the east. To compare the atmospheric and oceanic measurements, these were homogenized in time by taking the daily averages of each observation.

Heat fluxes were computed as follows, sensible heat ($Q_h$ in $\frac{W}{m^2}$) was based on Gill (1982) and Talley et al. (2011):

$$Q_h = \rho_a C_p C_h wVel \left[ SST - \left( T_{air} + \left( \frac{9.8}{1000} \right) z_{air} \right) \right] \tag{3}$$

where $\rho_a$ is the air density (in $\frac{Kg}{m^3}$), $C_p$ is the specific heat capacity of air at constant pressure (in $\frac{J}{(Kg*K)}$), $C_h$ is the bulk sensible heat transfer coefficient, $wVel$ is the wind velocity (in $\frac{m}{s}$), $SST$ is the Sea Surface Temperature (from satellite remote sensing in °C), $T_{air}$ is the air temperature, and $z_{air}$ is the height where the air temperature was taken (~10 meters above sea-level). $C_h$ was made dependent on the wind velocity and $\Delta SST = SST - T_{air}$ according to table values by Smith (1988).

Latent heat ($Q_e$ in $\frac{W}{m^2}$) was estimated following Gill (1982) and Castro *et al.* (1994):

$$Q_e = \rho_a C_e wVel L_v (q_s - q_a) \tag{4}$$

where $C_e$ is the exchange coefficient, $wVel$ is the wind velocity (in $\frac{m}{s}$), $q_s$ is the saturation specific humidity of the sea surface, $q_a$ is the specific humidity of air and $L_v$ is the latent heat of evaporation (in $\frac{W}{m^2}$). $C_e$ was made dependent on the wind velocity

and $\Delta T$ according to table values by Bunker (1976), whereas the specific humidity saturation of the sea surface ($q_s$) was calculated with (Gill, 1982):

$$q_s = \frac{0.62197 e_w}{(P_{atm} - 0.378 e_w)}$$ (5)

where $e_w$ and $q_a$ were explained above. The latent heat of evaporation ($L_v$) was measured as,

$$L_v = 2.5008 \times 10^3 - 2.3 SST$$ (6)

Moreover, from the weather station data, the impact of the hurricane winds on the water column was analyzed by computing the Ekman surface velocity ($U_E$) and the Ekman layer depth ($D_e$). $U_E$ was estimated following Rio et al. (2014):

$$U_E(z) = \beta(z)\tau e^{(i\theta(z))}$$ (7)

where the parameters $\beta$ and $\theta$ at the surface are $\beta(0)$=0.61 and $\theta(0)$=30.75, respectively. $\tau$ is the surface wind stress (in $\frac{N}{m^2}$):

$$\tau = C_d \rho_a wVel^2$$ (8)

$C_d$ is the drag coefficient, $wVel$ is the weather station wind velocity (in $\frac{m}{s}$), and $\rho_a$ is the air density defined above. The Ekman layer depth (in m) was based on Cushman-Roisin & Beckers (2011):

$$D_E = 0.4 \frac{u^*}{f}$$ (9)

where $f$ is the Coriolis parameter estimated at the position of the ADCP, $u^*$ is the turbulent velocity:

$$u^* = k \frac{cVel}{log\left(\frac{z}{z_0}\right)}$$ (10)

$cVel$ (in $\frac{cm}{s}$) is the vertically averaged current velocity, $k$ is the von Karman constant (0.41), $z$ is the mean current measurement depth (5.5 m in our case), and $z_0$ is the size of the ripples or gravel on the seafloor (~0.05 m).

### 3.2.3 Satellite imagery

For the analysis of (1) Surface winds, (2) Sea Surface Salinity (SSS), and (3) Sea Surface Temperature (SST) the following remote sensing images were used:

(i) Metop/ASCAT scatterometer: 1/4° Daily wind and wind stress maps from the Centre de Recherche et d'Exploitation Satellitaire (CERSAT), at IFREMER, Plouzané (France). More details on the data, objective, method, and computation algorithm are found in Bentamy and Croizé-Fillion (2012). Data and documentation are freely distributed at ftp://ftp.ifremer.fr/ifremer/cersat/products/gridded/MWF/L3/ASCAT/Daily/

(ii) The L3_DEBIAS_LOCEAN_v5. 1/4° 4-day Sea Surface Salinity maps (Boutin et al., 2018) have been produced by LOCEAN/IPSL (UMR CNRS/UPMC/IRD/MNHN) laboratory and the ACRI-st company that participate in the Ocean Salinity Expertise Center (CECOS) of the Centre Aval de Traitement des Donnees SMOS (CATDS). This product is distributed by the Ocean Salinity Expertise Center (CECOS) of the CNES-IFREMER Centre Aval de Traitement des Donnees SMOS (CATDS), at IFREMER, Plouzane (France) (ftp://ext-catds-cecos-locean:catds2010@ftp.ifremer.fr/).

(iii) The NOAA 1/4° daily Optimum Interpolation Sea Surface Temperature (Reynolds et al., 2002) was provided by the NOAA/OAR/ESRL PSD, Boulder, Colorado, USA, at https://www.esrl.noaa.gov/psd/.

Surface winds, SSS, and SST observations were spatially interpolated on a 25 km radius from the ADCP location, salinity and temperature data were later transformed to conservative temperature and absolute salinity to estimate $\sigma_\theta$ (in $\frac{Kg}{m^3}$), based on the thermodynamic equation of seawater TEOS-10 (McDougall & Barker, 2011). The precipitation brought by these storms caused changes in the seawater density ($\sigma_\theta$ in $\frac{Kg}{m^3}$), which were inspected employing the SSS and SST data. Whereas the remotely sensed surface wind stress was used to estimate the Ekman pumping ($W_E$ in m s$^{-1}$), following the formulas proposed by Cushman-Roisin & Beckers (2011):

$$W_E = \frac{1}{\rho_0}\left[\frac{d}{dx}\left(\frac{\tau^y}{f}\right) - \frac{d}{dx}\left(\frac{\tau^x}{f}\right)\right] \tag{11}$$

where $W_E$ (in $\frac{m}{s}$) is the vertical velocity estimated from the wind stress components ($\tau^x, \tau^y$) of the satellite surface winds near the ADCP location, $f$ is the Coriolis parameter, $\rho_0$ (in $\frac{Kg}{m^3}$) is the water density. Multiplying the Ekman pumping by the Ekman layer, the vertical advective transport ($W_{aT}$ in m$^2$ s$^{-1}$) was obtained:

$$W_{aT} = W_E D_E \tag{12}$$

### 3.2.4 Beach surveys

The most notable coastal impacts associated with the passage of storms are beach erosion and flooding. Beach profiles were employed to estimate the subaerial beach volume change by integration of beach elevation ($z \geq 0$ m) with respect to the cross-shore distance. The subaerial beach volume change can be readily obtained by subtracting the pre-storm from the post-storm beach survey. On the other hand, shoreline position was obtained by tracking the cross-shore location corresponding to $z = 0$ m for each transect, hence shoreline change was estimated as the difference between the pre- and post- storm shoreline location. To estimate the coastal flooding during the peak of the storm, beach elevation changes between subsequent surveys were employed as a proxy for the maximum water levels at each transect.

### 3.2.5 Video camera system

Time exposure images were employed to observe the storm effects on macrophyte wrecking, dune vegetation, and to estimate post-storm inundated areas. To quantify the impact in meters, image pixel coordinates were transformed to UTM coordinates by relating Ground Control Points (GCPs) to pixel positions (Simarro et al., 2017). Unfortunately, the strong winds moved the orientation of the cameras and even the GCPs. To solve this, a previous image with known GCPs was used as a reference to stabilize the images of interest (Arriaga et al., 2022). Following the methodology of Rutten et al. (2021) to detect Sargassum on images, the Support Vector Machine (SVM) algorithm is employed to detect the coverage of wrack, vegetation, and flooded areas (October 1, 2, 5, 20).

## 3.3 Numerical modeling

Two different numerical approaches were implemented in the study area. The numerical models were forced with different wind information and grid resolution. The first approach aimed to forecast the atmospheric and oceanic conditions generated during the pass of the events, while the second approach aimed to determine the wave and storm surge hazards created by the resulting waves and storm surge. A description of each modeling approach and its implementation is provided below.

### 3.3.1 Forecast modeling

The numerical models WRF, WWIII, and ADCIRC were employed to forecast nearshore hydrodynamics as described below.

#### 3.3.1.1 WRF model

The Weather Research and Forecasting model (WRF V.3.9), developed by the National Center for Atmospheric Research (NCAR), is characterized by being compressible, non-hydrostatic, with terrain-following hydrostatic pressure vertical coordinates and Arakawa-C horizontal grid staggering (Arakawa and Lamb, 1977). The model used the Runge-Kutta $2^{nd}$ and $3^{rd}$ order time integration schemes and the $2^{nd}$ to $6^{th}$ order advection schemes in both the horizontal and vertical. Moreover, it also uses a small time-split small step for acoustic and gravity-wave modes. For further details, refer to Skamarock et. al. (2008). The operational forecasting system established at the Atmospheric Sciences and Climate Change Institute (ICAyCC) at the UNAM (Ocean-Atmosphere Interaction Group, 2020) was used. The physics model parameterizations are the Kain-Fritsch cumulus parameterization scheme (Kain, 2004), the RRTM (Rapid Radiative Transfer Model) scheme for longwave radiation (Mlawer, et al., 1997), the Dudhia scheme for shortwave radiation (Dudhia, 1989), and the Yonsei University (YSU) scheme for the boundary layer (Skamarock et al., 2008; Hong, et al., 2006). In addition, the 5-layer thermal diffusion Land Surface Model (LSM) was used. This scheme, although simple, is adequate for most mesoscale studies and estimates the energy balance at a low computational cost (Dudhia, 1996). The Land Use and Land Cover soil category map data used were obtained from the USGS database with 24 classes (Loveland et al., 2000). Here, the forecast employed two one-way nested computational domains. The first domain (D01) has a 15 km horizontal grid resolution and includes Mexico, the GoM, part of the Caribbean Sea, and part of the central Pacific. The second domain (D02) has a resolution of 5 km and includes the central part of the Mexican territory. The forecast employed 30 vertical levels in a log-normal distribution, with the top of the atmosphere fixed at 50 mbar. The model equations were integrated every 120 s. For the initial and boundary conditions, the numerical model was initialized with the Global Forecast System (GFS) model at 0000 UTC data, every six hours with a one-degree spatial resolution. The operational system produces a 5-day forecast, however for this work only the 24-hour, 48-hour, and 72-hour forecasts were considered. It is known that a tropical cyclone's track and intensity forecast degrades rapidly, so it

is not reliable beyond 72 hours. Despite the performance of the forecasts being acceptable for this particular case (see Table 1), it was not considered for discussion. The correlation coefficients halved for wind, waves, and sea level parameters beyond with respect to 72-hours.

### 3.3.1.2 Wave Watch III

The WAVEWATCH III (WWIII V.5.16) model is a third-generation model developed by the NOAA (NCEP), characterized for solving the random phase spectral action density balance equation for wave number direction spectra. The implicit assumption of this equation is that properties of the medium (water depth and current) and the wave field itself, vary over time and space scales that are much larger than the variation scales of a single wave. This model also considers options for extremely shallow water (surf zone), as well as wetting and drying of grid points. The total wave energy and the local and instantaneous
spectrum of the waves can be obtained as model outputs, where the latter can be reduced to a two-dimensional function. The parameters of significant wave height, mean period, and direction of propagation are also model outputs. The parameterizations of the operational forecasting system used in this work are: the Cavaleri and Rizzoli (1981) term to represent the linear growth of the waves and the parameterization described by Tolman and Chalikov (1996) in the terms that define the integral growth of waves. To represent nonlinear processes, the Discrete Iterations Approximation described by Hasselmann et al. (1985) was
used. The bottom friction was represented with an empirical linear function described in Hasselmann et al. (1973). The forecast uses a rectangular, rectilinear grid. The model was implemented in two one-way nested computational domains: the World Ocean, on one side the Pacific Ocean, and on the other, the Gulf of Mexico. Atmospheric forcing was obtained from the Global Forecast System (GFS from NCEP; https://www.ncdc.noaa.gov/data-access/model-data/model-datasets/global-forcast-system-gfs) and the Weather Research and Forecasting model (WRF), respectively for each domain. The bathymetry used was
the ETOPO1 elevation database from the National Centers for Environmental Information (NOAA; https://www.ngdc.noaa.gov/mgg/global/), with a spatial resolution of one arcminute. The operational system produces a 5-day, 3-hourly forecast; however, only the 24-hour, 48-hour, and 72-hour forecasts were considered for this work.

### 3.3.1.3 ADCIRC

The ADvanced CIRCulation Model for Oceanic, Coastal and Estuarine Waters (ADCIRC V.52.30.13 with NetCDF files
support) is a system of programs for solving time-dependent free surface circulation and transport problems in 2D and 3D. These programs solve the movement equations for a rotating fluid through the Boussinesq and hydrostatic pressure approximations, both discretized in space by the finite element method and in time by the finite difference method. This way of solving the movement equations allows the use of highly flexible unstructured grids. ADCIRC calculates the surface elevation from the Generalized Wave-Continuity Equation (GWCE) and the current velocity from the momentum equations.
All nonlinear terms have been retained in these equations. Its applications include wind and tidal circulation modeling, flood and storm surge analysis, dredging and disposal feasibility studies, larval transport studies, as well as for nearshore marine

operations. The configuration used in this work has a resolution equal to or less than 500 m along the Mexican coast, reducing the resolution to 4 km for the northern Gulf of Mexico. Along the open boundary, eight harmonic tidal constituents are prescribed (M2, S2, K2, N2, K1, O1, P1, and Q1) obtained from TPX0 (tpxo.net/global; Egbert et al., 2002). Surface forcings,
including hourly winds and sea level atmospheric pressure, were taken from the WRF atmospheric model described in the previous section. Initial conditions were obtained from the atmospheric model output after regridding. Table 2 summarizes the setup for each of the aforementioned models.

Table 2. Setup characteristics for the WRF, WWIII, and ADCIRC models.

| Numerical model | Computational domain | Temporal resolution | Spatial resolution | Model outputs |
|---|---|---|---|---|
| WRF | 74ºW -123ºW 4ºN - 38ºN | 1 hr | 15 km | Surface wind velocity (km/h), wind direction (º), air temperature (ºC), reduced pressure at sea level (hPa) |
| WWIII | Gulf of Mexico | 1 hr | 27.8 km | Significant wave height (m), wave period (s), and peak wave direction (º) |
| ADCIRC | 80ºW - 99ºW 16ºN - 31ºN | 1 hr | 0.5 km | Sea level anomaly (m) |


### 3.3.2 Hazards assessment modeling

The numerical model MIKE 21 HD FM (hydrodynamic) and MIKE 21 SW (waves) were employed to assess wave and storm surge conditions. The MIKE 21 SW is a third-generation spectral wave model which resolves the wave action equation employing non-structured grids using a finite volume. The MIKE 21 HD FM (DHI, 2017) solves the RANS equations with
the Boussinesq and hydrostatic pressure approximations. The numerical model employs flexible meshes, allowing the resolution in the area of interest to be increased using a finite volume. This numerical model assumes a Coriolis force, baroclinic density, eddy viscosity of 0.28 based on the Smagorinsky formulation, and constant wind friction coefficient (0.001255 for $wVel$ <7 m/s and 0.002425 for $wVel$ > 7 m/s). The MIKE 21 SW is implemented in stationary mode with a logarithmic discretization in the frequency domain and directional spectra divided into 32 bins. The wave and hydrodynamic
models employed a computational domain covering the Gulf of Mexico with an 8-km resolution near the coast. Moreover, the resolution in the hydrodynamic model further increases in coastal areas up to 40 m in the area of interest. The topographic

data used were obtained from a 1 m resolution LiDAR survey from 2011 of the Yucatan coast, while the bathymetry was obtained from local surveys complemented with ETOPO 1 data (Amante and Eakins, 2009).

## 4 Results

### 4.1 Hurricanes Gamma and Delta

#### 4.1.1 Atmospheric conditions

The passage of Hurricane Gamma induced an increase in the wind speed at Sisal, Yucatan, reaching a peak magnitude of 20 m s$^{-1}$ on October 3, 2020, and maintaining sustained winds around 15 m s$^{-1}$ for the following days until October 6 (Figure 2a). The wind direction switched from the NE to the NNW on October 4 to 6. The wind velocity dropped to 5 m s$^{-1}$ on October 6, increasing suddenly to 20 m s$^{-1}$ on October 7 due to the passage of Delta. A sustained drop followed the wind increase throughout the same day due to Delta's fast translation speed. Wind direction switched from the NW to the SW as the storm passed. The atmospheric pressure (Figure 2b) shows a significant decrease below 1000 mbar during the passage of Delta. The atmospheric temperature remained relatively steady, around 25 ºC, and the diurnal variability associated with the sea breeze recovered after October 8 (Figure 2c). Heavy rain (50 mm) fell during October 2 followed by a lower peak on October 7 (Figure 2d).

#### 4.1.2 Oceanic conditions

In situ measurements at 11-m water depth allowed us to assess the effects of hurricanes Gamma and Delta on the nearshore sea. The ADCP time series spanned more than three years of data. However, Figure 3 focuses on the daily averages of different variables from July to October 2020, to highlight the effects of hurricanes Gamma (October 5) and Delta (October 7). Figure 3a shows the wind velocity and the sea surface height oscillation before, during, and after the passage of such atmospheric events. The winds accelerated to reach 14.4 m s$^{-1}$ (52 km h$^{-1}$), blowing from the north and promoting a sea-level set up ($\Delta\eta$) of 9 cm associated with Gamma and a 30 cm set up during Delta. Ocean currents increased to reach the maximum value registered in the time series since 2018 (59 cm s$^{-1}$) during Gamma (Figure 3b).

Furthermore, we analyzed the influence of wind stress over the nearshore sea by looking at the Ekman layer depth ($D_E$) and the surface currents (Ekman currents, $U_E$). The $D_E$ average value for the ADCP location is 70$\pm$46 m (estimated from time series since 2018), that is, the Ekman layer commonly encompasses the entire water column. In general, regional surface winds can generate an Ekman layer depth greater than 10 m, 92 % of the time. During the passage of Delta and Gamma, the Ekman layer depth attained a mean value of 169 m and reached a maximum value of 350 m, implying significant water mixing throughout the water column and sediment resuspension due to seabed friction. $U_E$ accelerated to the maximum value (26 cm s$^{-1}$) during Gamma. A comparison between the maximum current magnitude and the maximum currents due to the surface

wind stress ($U_E$), suggests that the latter represented 44 % of the current magnitude (Figure 3c). The vertical advective transport ($W_{aT}$) generated by Gamma and Delta (time series not shown) is limited by the depth of the study region, which restricts the Ekman layer depth. The vertical advective transport showed an average positive value during Gamma ($1.6 \times 10^{-4}$ m$^2$ s$^{-1}$), i.e. an upwelling transport, and a sudden change during Delta from positive to negative values, changing from upwelling to downwelling transport in 2 days (from $4.0 \times 10^{-4}$ to $-6.2 \times 10^{-4}$ m$^2$ s$^{-1}$). At the same time, Gamma and Delta promoted a drop in air ($T_{air}$) and sea bottom temperature ($T_b$) of 5 and 3 °C, respectively (Figure 3d). Moreover, the accumulation of freshwater due to the high precipitation volumes induced a decrease in the sea surface density, exhibiting low values (1022.7 kg m$^{-3}$) at the end of these storms (Figure 3e).

To investigate the heat exchange between the sea surface and the atmosphere, we estimated the sensible and the latent heat fluxes (Figure 3f). The sensible heat ($Q_h$) is a proxy for the heat gain or loss from the sea surface due to thermal gradients between this and the adjacent air. On the other hand, the latent heat ($Q_e$) represents the heat exchange ascribed to evaporation/condensation processes between the sea surface and the atmosphere. A positive (negative) value represents a heat output (input) from the sea. Figure 3f shows the maximum sensible heat loss during the passage of Gamma. Thus, field observations suggest that both Gamma and Delta absorbed heat from the sea, Delta to a lesser extent. The latent heat ($Q_e$) reached a maximum value during Gamma, but was also very high during Delta, showing a large amount of seawater evaporation, or vapor condensation (cloud formation) in the atmosphere for both events. Moreover, high values observed at the end of the time series (i.e., 25/10/2022) could be associated with the high air moisture caused by the floods left by the hurricanes. Ambient moisture condenses into raindrops, forming clouds and taking latent heat from the ocean's surface. Therefore, cloud formation could be the cause associated with the higher $Q_e$ values observed at the end of this time series.

Table 3. Daily mean and maximum values during Gamma and Delta.

| Event | wind speed (m s$^{-1}$) | | wind stress (N m$^{-2}$) | | $D_E$ (m) | | $U_E$ (m s$^{-1}$) | | $W_{aT}$ ($10^{-4}$ m$^2$ s$^{-1}$) | | Hs (m) | | Rain (mm) | |
|---|---|---|---|---|---|---|---|---|---|---|---|---|---|---|
| | mean | max | mean | max | mean | max | Mean | max | mean | max | mean | max | mean | max |
| Gamma | 9.4 | 14.4 | 0.2 | 0.4 | 176 | 350 | 0.12 | 0.23 | 1.6 | 3.1 | 1.4 | 1.9 | 22 | 54 |
| Delta | 6.4 | 11.4 | 0.1 | 0.2 | 154 | 294 | 0.07 | 0.12 | -0.9 | 4.0 | 1.3 | 1.7 | 11 | 40 |

An Empirical Orthogonal Function analysis (EOF) was performed on the alongshore component ($u$) of the ADCP currents, for the three years of the time series. Figure 4 shows this result from July to October, 2020. Mode 1 accounted for 93 % of the explained variance; the time distribution depicted the strongest and fastest current fluctuation during the passage of these storms (orange lines in panel a1), where negative (positive) values represent a westward (eastward) flow during Gamma (Delta). The spatial distribution (Figure 4.a2) revealed a typical bottom Ekman layer distribution for the alongshore current, with a vertical shear ($\frac{\Delta u}{\Delta z}$) of 0.005 s$^{-1}$, that is an alongshore current difference of 4 cm s$^{-1}$, in the 8 m spanned by the ADCP. During the passage of Gamma and Delta, $\frac{\Delta u}{\Delta z}$ presented a mean value of 0.024 s$^{-1}$, and a maximum of 0.077 s$^{-1}$. Figure 4b illustrates the alongshore current time-depth distribution. Coastal currents off Sisal flow preferentially towards the west, from the Caribbean Sea towards the Gulf of Mexico, 80 % of the time, considering the three-year time series. However, it is common to find sporadic eastward flows, as shown by the red colors. During Gamma and Delta, the great mixing capabilities of these cyclones were evidenced by the strong and fast current fluctuation promoted in the water column over a few days (dashed lines in Figure 4b).

Intense winds drove energetic waves ($H_s > 2$ m) during the passage of both Gamma and Delta (Figure 5a). Gamma induced NNW waves higher than 1.5 m and $T_p > 6$ s from October 3 to October 6. Wave energy decreased during October 7 but increased by the end of the same day due to the passage of Delta, reaching $H_s > 2$ m for a few hours due to the fast translation speed. Wave direction was from the NW (Figure 5c) and the peak period was around 8 s (Figure 5b). Therefore, the alongshore sediment transport is expected to occur in the eastward direction. After the passage of the tropical systems, the typical low-energy wave conditions associated with sea breezes were restored. Mean sea level at the coast increased 0.2 and 0.3 m for Gamma and Delta, respectively (Figure 5d).

## 4.2 Coastal impacts

### 4.2.1 Wrack and vegetation

Vegetation on the subaerial beach profile and foredune was present in Sisal before tropical storms Gamma and Delta (Figure 6a,7a). The more consolidated and dense vegetation was present 40-m away from the shoreline and remained unaltered after the meteorological events. However, high-water levels associated with the passage of Gamma (October 2-5) either damaged or buried the beach vegetation in the low-elevation area in the vicinity of the port jetty and the central beach region (Figure 6b,7a). The further increase in the water levels and wave energy during the passage of Delta (October 8) affected all the pioneer vegetation on the beach's shoreward-most location (Figures 6d).

Prior to the storms, a small quantity of wrack was present along the beach (Figures 6a,7b). The increase in the incoming wave energy induced significant sediment transport and seabed erosion on the nearshore, causing the dislodgement of seagrasses.

The seagrass was transported onshore by waves and currents and wrecked on the beach face, especially west of the pier (Figures 6c,7b). After the storms, the sea breezes re-distributed the seagrass along the coast, accumulating most of the wrack east of the jetty (Figures 6d,7b).

### 4.2.2 Beach morphology and flooding

Beach profiles undertaken before and after Delta and Gamma allow assessing the storm impact on the beach morphology and estimating flooding. Pre- and post-storm beach profiles were analyzed to determine beach changes. The largest changes in both shoreline position and subaerial beach volume occurred in the vicinity of the pier and the port's jetty (see P02-P03 and P20-P22 in Figure 8). The shoreline position east of the port (P20) and the pier (P02) retreated 22 m and 10 m, respectively. On the other hand, 15 m and 2 m shoreline advances were observed west of the jetty (P21) and the pier (P03). The mean
shoreline change between transects P01 and P20 was -3 m, with transects P15-P18 showing a net increase. The latter suggests that significant eastward alongshore transport occurred during the storm sequence, induced by the NNW waves (Figure 5c), redistributing the existing sediment east of the port (P19-P20) to adjacent transects (P15-P18). West of the port of Sisal (transects P21-P40) the mean shoreline advance was 4 m, with transects P27, P30, P31, P33, and P34 presenting a shoreline retreat in this area (Figure 8a). The shoreline advance along P37-P40 seems to be related to the formation of a 0.20 m berm. It
is important to point out that beach scarp erosion contributed to beach sediment accumulation at some transects.

The subaerial beach volume change, associated with cross-shore transport, presented an overall net increase (Figure 8 c-d). A lower impact on beach volume was observed at transects P04-P10 located in the area where significant seagrass wrack occurred. The sequence of storms did not induce a significant mean subaerial volume change (0.8 $m^3$/m), and the maximum
volume increase (+11 $m^3$/m)/decrease (-18 $m^3$/m) corresponded to transects located west/east of the jetty (Figure 8c-d). Significant volume losses also occurred at P02, P24, and P27. Away from the structures, significant sediment supply by the storms contributed to an increase in beach elevation (e.g., Tuck et al., 2021).

Beach morphology changes are also employed as a proxy for coastal flooding on the beach located in front of the coastal community of Sisal (i.e., P01-P20). Bed elevation increase was observed landward of the shoreline at most profiles (red and
orange areas), whereas erosion was maximum east of the structures (blue areas) (Figure 9a). The landward limit of observed bed changes was used as a proxy for the maximum horizontal swash excursion $X_{max}$ (Figure 9b), showing alongshore differences with a maximum closer to the port's jetty. The bed change associated with the maximum $z$ was used as a proxy for the maximum water levels ($Z_{max}$=tide + storm surge + runup), implying that swash flows reaching that area were significant enough to induce sediment transport. The maximum elevation of such changes was found at P19 and the minimum at P07,
corresponding to elevations of 1.7 m and 0.5 m, respectively (Figure 9c). The lower values are correlated with the areas that presented wrack coverage during the storms (Figure 7b).

The most vulnerable area to floods is located east of the jetty. The heavy rain of October 2 was capable of flooding a 200 m stretch of beach (Figures 6b,7c) and the succeeding forcings of Gamma propagated the flooded area farther to the east (Figures 6c,7c). The high vulnerability is due to dredging practices that have created a low-lying zone at this location.


### 4.2.3 Coastal aquifer

Monitoring wells W4, W5, and W7a located 20 km, 5 km, and 200 m from the coast, provide information on the oceanic and terrestrial forcing on the coastal aquifer. Figure 10 shows the relative levels of the hydraulic head at each well and the precipitation at Sisal. Wells W7a, and W5 show the diurnal tidal modulation on the hydraulic head (Figure 10a). All wells
show an increase in the water table owing to the recharge following the storms. This is more evident at the well located 20 km from Sisal (W4) which shows an increase of more than 2 m following the passage of Gamma and Delta, reaching a maximum level on October 7. It is worth noting that coastal wells W5 and W7a do not show such an increase in the water table due to confined aquifer conditions that do not allow rapid infiltration of the precipitation (Figure 10b); however, the storm effects can be seen in the change in amplitude (decrease) of the tide caused by the increase in aquifer discharge. The southern limit of
such confinement is not well known, and hence back-barrier flooding might occur when the water table exceeds the confinement level south of the aquifer, preventing the hydraulic head in well W7a and W5 from increasing further (Perry, 1989; Pino 2011).

### 4.3 Numerical modeling

### 4.3.1 Forecast modeling

Figures 11 and 12 show the time series of the measured and forecast data for 24, 48, and 72 hrs. Figure 11 shows that the 24-hour forecasts adequately represent the variability of air temperature, wind direction, and atmospheric pressure. In the case of air temperature (Figure 11a), the forecasts underestimate the diurnal variability, while during the storm events, they fit appropriately, thus, a relatively low correlation coefficient was obtained. For the wind speed, the forecasts consistently underestimate the magnitude of the wind (Figure 11b), mainly during extreme events; however, the wind direction presents
less bias than its magnitude (Figure 11c). Regarding atmospheric pressure (Figure 11d), the 24-hour forecast shows a significantly high correlation coefficient, which suggests high reliability. Both the 48-hour and 72-hour forecasts fail to describe the atmospheric conditions during these events.

In the same way as the atmospheric variables, the significant wave height, wave direction, and mean sea level were analyzed for the same period. Figure 12a shows that the forecasts adequately represent the temporal variability of the wave energy but
significantly underestimate the significant height most of the time. The forecast that best fits the significant wave height variability is that of 24 hours, mainly during Gamma. However, despite presenting a high correlation coefficient, the forecast

significantly underpredicts the significant wave height during Delta. Regarding the wave direction, the three forecasts present an average bias of 1.6º (Figure 12b). Analyzing the change in sea level due to the approach of storms to the coast of Sisal, Figure 12c shows that all the forecasts overpredict and underpredict the mean sea level during Gamma and Delta, respectively. The 24-hour forecast is the one that best reproduces the sea level variability of the measured signal, however during extreme events, the 72-hour forecast has a lower RMSE and bias concerning the measured data than the 48-hour forecast. Table 4 shows the statistical fit parameters that support the results obtained and, additionally, Table 5 shows the correlation coefficient (CC), the root-mean-square error (RMSE), and the bias (BIAS) of each measured variable with respect to the forecasts.

Table 4. Statistical parameters at the 5% significance level. df: degree of freedom; STD: standard deviation; ci: confidence interval.

| Variables | t-test | df | STD | p-value | CI | |
|---|---|---|---|---|---|---|
| Temperature | 13.17 | 4 | 0.08 | 0.0001 | 0.35 | 0.54 |
| Wind Speed | 6.29 | 4 | 0.20 | 0.003 | 0.32 | 0.82 |
| Wind Direction | 2.96 | 4 | 0.25 | 0.042 | 0.02 | 0.64 |
| Atmospheric Pressure | 6.14 | 4 | 0.22 | 0.004 | 0.33 | 0.88 |
| Significant Wave Height | 6.01 | 4 | 0.24 | 0.004 | 0.34 | 0.93 |
| Wave direction | 4.99 | 4 | 0.22 | 0.008 | 0.22 | 0.75 |
| Sea Level | 3.12 | 4 | 0.24 | 0.036 | 0.04 | 0.64 |

Table 5. Representative statistics that validate the numerical simulations of the forecasts of the WRF, WWII, and ADCIRC models. The correlation coefficient (CC), root mean square error (RMSE), and bias (BIAS) of the analyzed variables are shown. The standard deviation (STD) of the measured data is also shown.

| Variables | Units | 24 hrs forecast | | | 48 hrs forecast | | | 72 hrs forecast | | | Measured data |
|---|---|---|---|---|---|---|---|---|---|---|---|
| | | CC | RMSE | BIAS | CC | RMSE | BIAS | CC | RMSE | BIAS | STD |
| Temperature | [°C] | 0.58 | 1.95 | 0.75 | 0.40 | 2.21 | 0.68 | 0.39 | 2.18 | 0.58 | 1.7 |
| Wind speed | [ms$^{-1}$] | 0.81 | 4.01 | 3.06 | 0.69 | 4.25 | 2.97 | 0.56 | 4.82 | 3.33 | 4.2 |
| Wind direction | [°] | 0.66 | 1.25 | 0.05 | 0.50 | 2.27 | 0.21 | 0.24 | 2.00 | 0.80 | 100.4 |
| Atmospheric pressure | [mb] | 0.93 | 1.20 | 0.09 | 0.70 | 2.71 | 0.57 | 0.56 | 3.37 | 0.61 | 3.2 |
| Significant wave height | [m] | 0.94 | 0.26 | 0.19 | 0.77 | 0.43 | 0.21 | 0.64 | 0.52 | 0.28 | 0.5 |
| Wave direction | [°] | 0.67 | 3.23 | 1.30 | 0.63 | 3.47 | 1.44 | 0.58 | 3.39 | 2.05 | 144.1 |
| Sea level | [m] | 0.64 | 0.19 | -0.10 | 0.46 | 0.23 | -0.11 | 0.38 | 0.22 | -0.06 | 0.2 |

### 4.3.2 Hazard assessment

The assessment of the waves and storm surges generated by Gamma and Delta is shown in Figures 13-15. Regarding the waves generated by these events, Figure 13 shows the results of maximum wave height (Fig. 13a and 13c) and the maximum mean period (Fig. 13b and 13d) obtained during the entire path of Gamma (Fig. 13a and 13b) and Delta (Fig. 13c and 13d). As can be seen, the waves generated by the Delta event were much larger than Gamma (i.e., northern Yucatan Peninsula), indicating a greater hazard.

Figure 14 shows the hydrodynamic simulation results denoting the maximum attained storm surge levels for Sisal. Delta generated larger flood areas than Gamma, mainly affecting the settlements near the lagoon area. The hydrodynamic and wave models were initially run-in forecast mode using the National Hurricane Center (NHC) prediction to forecast hazard areas and alert the local authorities. The results presented herein are the post-event analysis based on the setup used in the forecast model. The models used were calibrated for the Gulf of Mexico based on historical events but not for the Yucatan coast specifically. The storm surge assessment with the actual storm tracks shows higher values for Delta as it was a stronger event and passed closer to the study area. Sisal was barely affected by Gamma, while Delta created flooding in the deposition area updrift of the harbor and on the lagoon side of the town. In this sense, more studies need to be carried out in order to have a calibrated and validated forecast model for the area.

# 5 Discussions

## 5.1. Coastal resilience

The knowledge of coastal resilience to hurricane events is important for coastal communities on barrier islands. We analyze the time series of the beach morphology and the water table to determine how long the impact of these events lasted at the study site. Figure 15a shows the spatio-temporal evolution of the subaerial beach volume at transects located in front of the coastal community (i.e., P01 to P20) during the year following the storms. Observed morphological changes are significantly smaller at beach profiles located away from the structures (P05 to P10). The beach evolution is strongly influenced during the

following months by the sediment impoundment by the port's jetty. Significant alongshore differences were observed in the subaerial beach evolution; hence, we present the time series at selected transects (Figure 15b). For instance, the transect in the vicinity of the jetty (P20 in Figure 15b) presented significant subaerial beach volume losses (20 $m^3/m$) after the passage of the hurricane events. The negative trend continues during the winter and reverses in the summer of 2021 without reaching the pre-disturbed condition. On the other hand, transect P16 shows an increase in the subaerial beach volume after the passage of the

two events and remained relatively stable during the following year, while transect P7 did not present a significant subaerial volume change and remained in the pre-disturbed condition (Figure 15b).

For the water table, the aquifer took about six months to reach the pre-disturbed conditions; this effect is more evident at well W7a. Previous studies have found similar results: the effects of hurricanes on the aquifer can last for months (Yam-Caamal and Graniel-Castro, 2014; Covacs et al. 2017; Kovacs et al. 2017). Some of those effects are beneficial for the population

because there is a greater recharge of the freshwater resource. Still, on the other hand, the flooding experienced inland (e.g. in the capital city) after hurricanes Gamma and Delta, is caused partially due to the storm drainage systems not being able to drain the stormwater efficiently (the water table is too high for the drainage to work properly); Canul-Macario (sub judice) estimated that this effect could last for more than five months.


## 5.2 Limitations

The present study focuses on obtaining high-resolution measurements that can help to calibrate numerical models to be further implemented in other areas. Nevertheless, the location of the hurricane's track can play an important role in the observed impact. Hence, concurrent measurements at different places along the northern Yucatan coast need to be considered in future

field efforts. Field observations suggest that our DGPS measurement errors are significantly smaller than the observed changes in the beach and hence do not affect the conclusions reached in the present study regarding beach morphodynamics. On the other hand, the confinement of the aquifer (Perry, 1989; Villasuso-Pino et al. 2011; Canul-Macario et al. 2020) plays an important role in the dynamics of the coastal aquifer. It is well known that the effects of the tide propagate further when

compared to unconfined aquifers (White and Roberts 1994; Canul-Macario et al. 2020). In the case of Yucatan, the confining layer boundaries are not well known, Villasuso-Pino et al. (2011) show that its width decreases eastward. Therefore, more research is required on the coastal aquifer confinement to fully understand the coastal aquifer dynamics. Also, the effects of hurricanes on the aquifer of Yucatan are not well studied. Canul-Macario et al. (2020) studied the propagation of the atmospheric and meteorological tide on the coastal aquifer and found that meteorological tides propagate further inland. More detailed data is required to understand how hurricanes may impact the position of the saline interface in the aquifer, for example, and the role of the free aquifer on coastal flooding during such events (e.g., Geng et al., 2021), highlighting the need to estimate compound flooding at this location.

The numerical modeling presents limitations in implementing the forcing and boundary conditions due to the lack of some terrestrial and atmospheric processes not considered. The spatial and temporal resolution for the wind field associated with the storm passage is not high enough to capture the hydrodynamic response near the coast for such events that move/travel offshore. The uncertainty in the topography and bathymetry is important for the model implementation since it controls wave transformation and coastal flooding. Moreover, field observations suggest that the flooding on the barrier island's lee side was associated with the high precipitation and the increase in the water table. These processes were not accounted for in the numerical modeling approach. Therefore, monitoring water levels in the wetlands would provide greater insights into the importance of precipitation in the flooding of the barrier island.

## 6 Conclusions

Tropical storms are important hazards on barrier islands of micro-tidal beaches along the northern Yucatan Peninsula. We investigate the impacts of hurricanes Gamma and Delta from the ocean to the coast using field observations and numerical models. Strong winds drive water mixing across an extensive area due to the shallow continental shelf. The study area is located more than ~ 200 km away to the west of the center of the two storms. Although their impact lasted less than a week over the water column, the influence of both events in the oceanographic region was notable. Moreover, energetic waves and coastal currents induced significant sediment transport in the nearshore and were responsible for disaggregating macrophytes (seagrass) to be further transported to the shore. The wracks' presence provided natural shoreline protection by increasing wave dissipation in shallow waters. Significant beach changes occurred due to the presence of coastal structures (e.g., port jetties), inducing alongshore sediment transport gradients. The beach located in the vicinity of the structure has a lower capability to recover from subaerial beach volume losses after the storms and did not reach the pre-storms condition after one year. Strong winds and low atmospheric pressure induced a storm surge in the order of the tidal range. The high-water levels affected beach vegetation but also increased the subaerial beach volume due to cross-shore sediment transport. On the other hand, heavy rain increased the water level in the wetlands. The confinement of the aquifer plays an important role in the dynamics of the coastal aquifer and took several months to return to the pre-storms water level. More detailed data is required to understand how hurricanes may impact the position of the saline interface in the aquifer, and the role of the free aquifer on

coastal flooding during such events (e.g., Geng et al., 2021), highlighting the need to estimate compound flooding at this location.

## Data availability

Data is available upon reasonable request.

## Author contribution

Conceptualization: ATF and GMM. Methodology: ATF, GMM, JK, RPC, JA, CMA, MEAA, GLF, JZH. Field data acquisition and analysis: ATF, GMM, JK, RPC, JA, JAG, GLF, JZH; Numerical modelling: MEAA, CMA, JZH. All authors contributed to the writing of the manuscript.

## Competing interests

The authors declare that they have no conflict of interest.

## Acknowledgments

Field support was provided by José López González, and Camilo Rendón Valdez. IT technical support was provided by Gonzalo Uriel Martín Ruiz. Numerical modelling support was provided by Pablo Ruiz-Salcines and Grupo Interacción
Océano-Atmósfera at the Instituto de Ciencias de la Atmósfera y Cambio Climático-UNAM. Financial support was provided by CONACYT through Investigación Científica Básica (Project 284819), Cátedras Program (Project 1146), Laboratorios Nacionales Program (Project LN 271544) and Infraestructura Program (INFR-2014-01-225561). Additional financial support was provided by PAPIIT DGAPA UNAM (Project IA101422). Tidal data was provided by the Servicio Mareografico Nacional and the meteorological data from the Red Universitaria de Observatorios Atmosféricos de la Universidad Nacional Autonoma
de Mexico (RUOA).

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

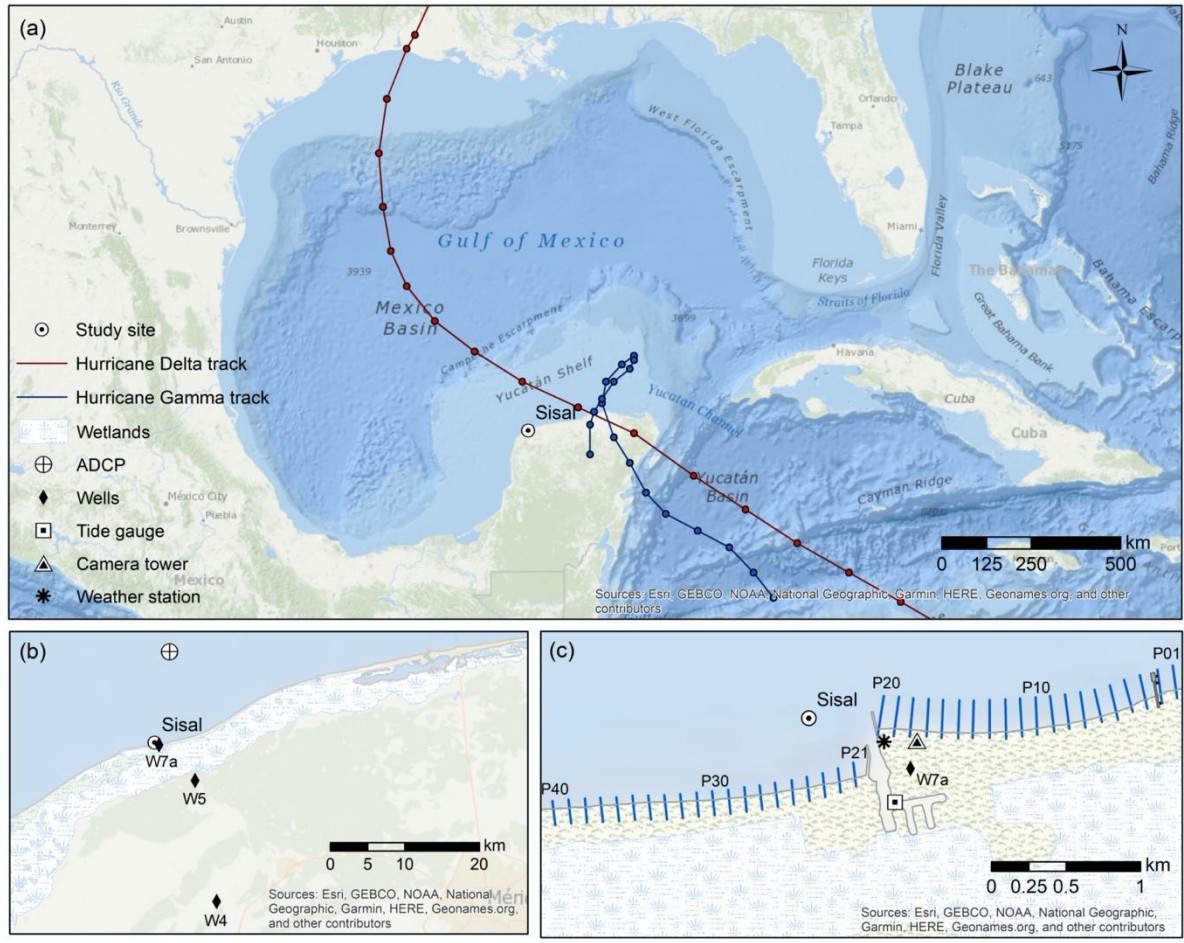

Figure 1: Study area showing: (a) best track positions for Hurricanes Gamma and Delta (National Hurricane Center, https://www.nhc.noaa.gov/data/tcr/index.php?season=2020&basin=atl); (b) the location of the ADCP, the monitoring wells; and (c) coastal monitoring systems and beach transects.

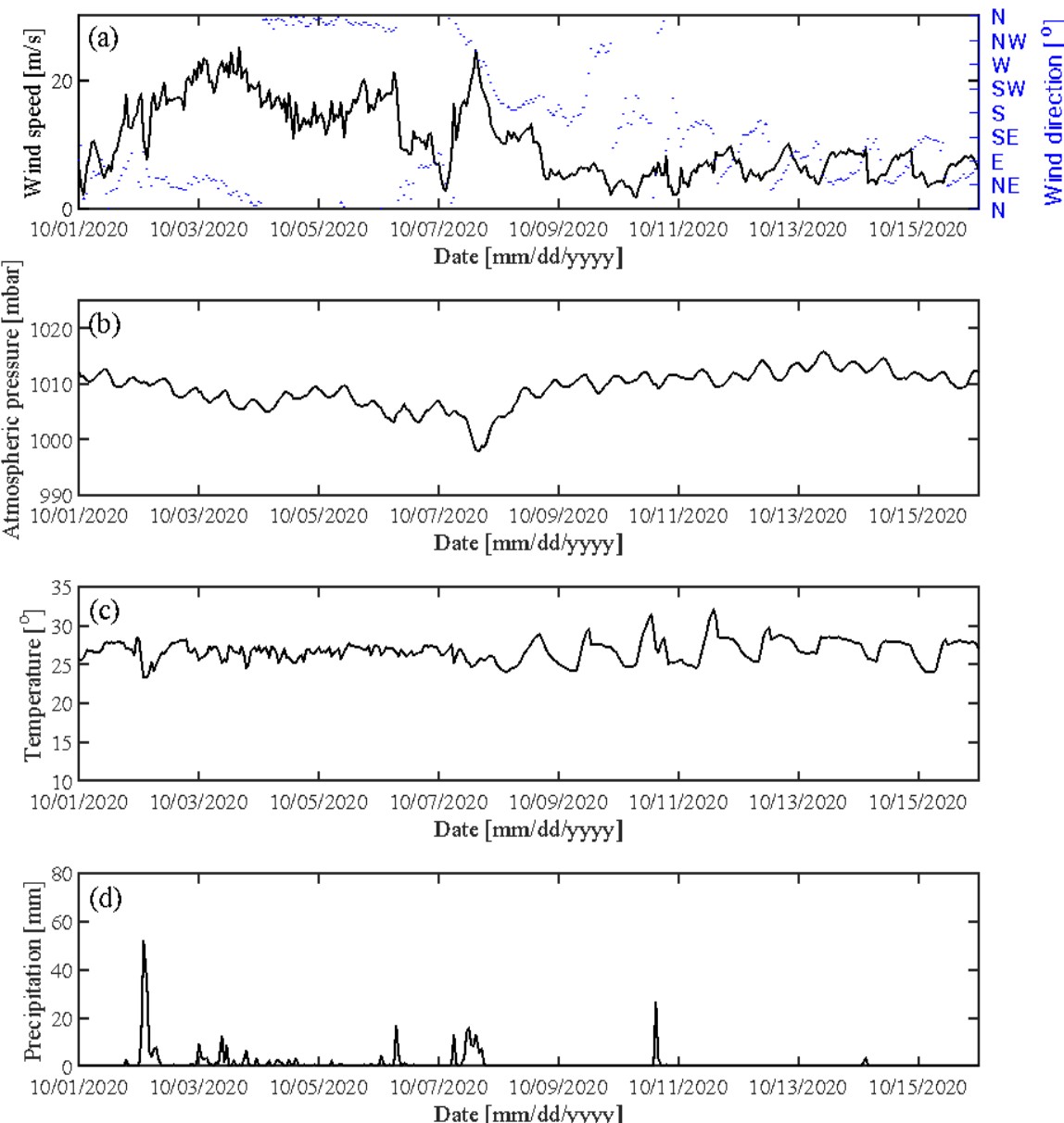

**Figure 2: (a) Wind magnitude and direction, (b) atmospheric pressure, (c) air temperature, and (d) precipitation measured by the weather station at Sisal, Yucatan.**



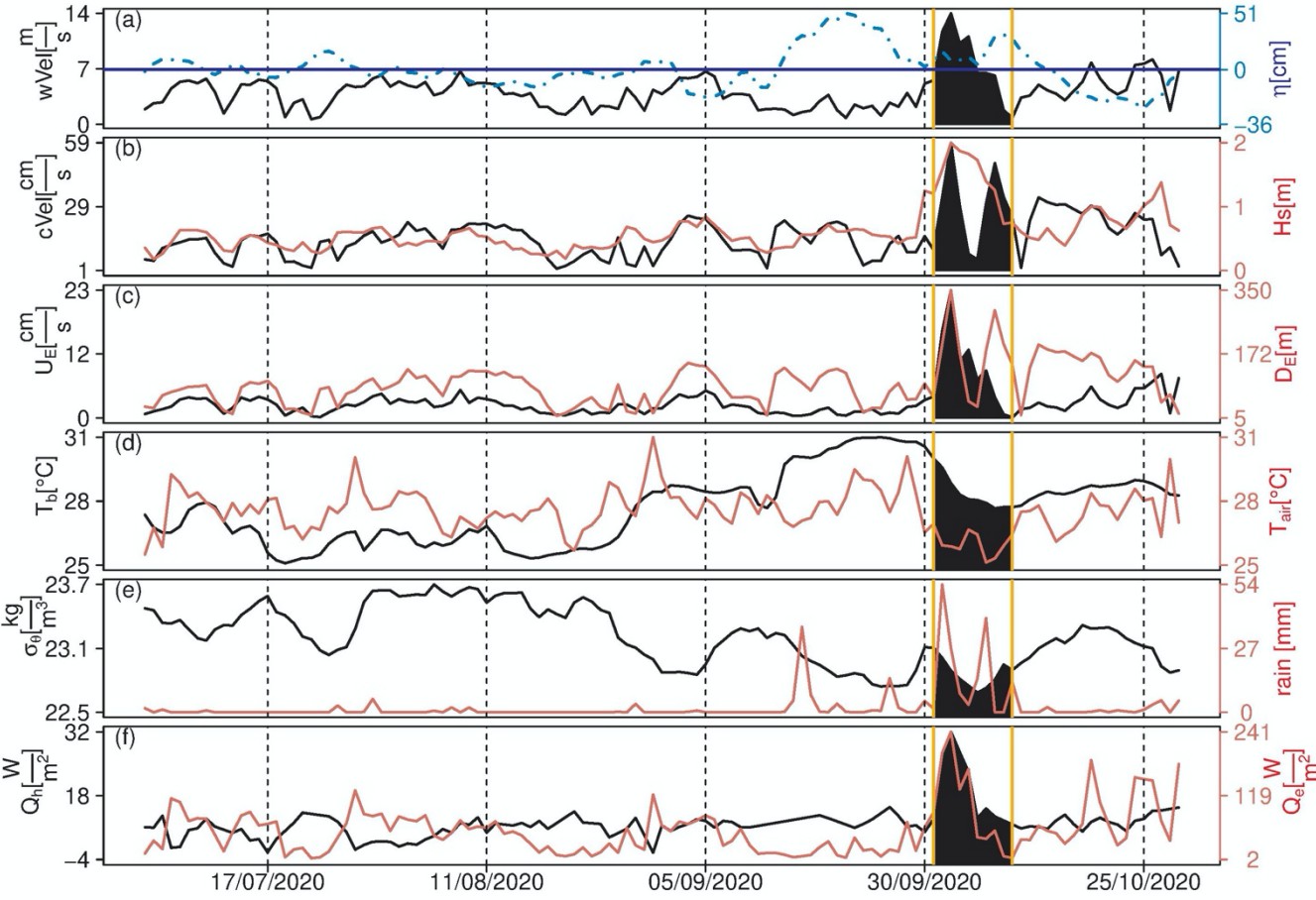

**Figure 3: Time series of (a) wind velocity and sea surface height (in blue), (b) current velocity and significant wave height (in red), (c) Ekman currents and Ekman layer depth (in red), (d) sea bottom temperature and air temperature (in red), (e) seawater density and precipitation (in red), and (f) sensible and latent heat (in red), estimated from the meteorological station, the moored ADCP and the satellite data. The dates of the passage of storms Gamma and Delta are highlighted.**


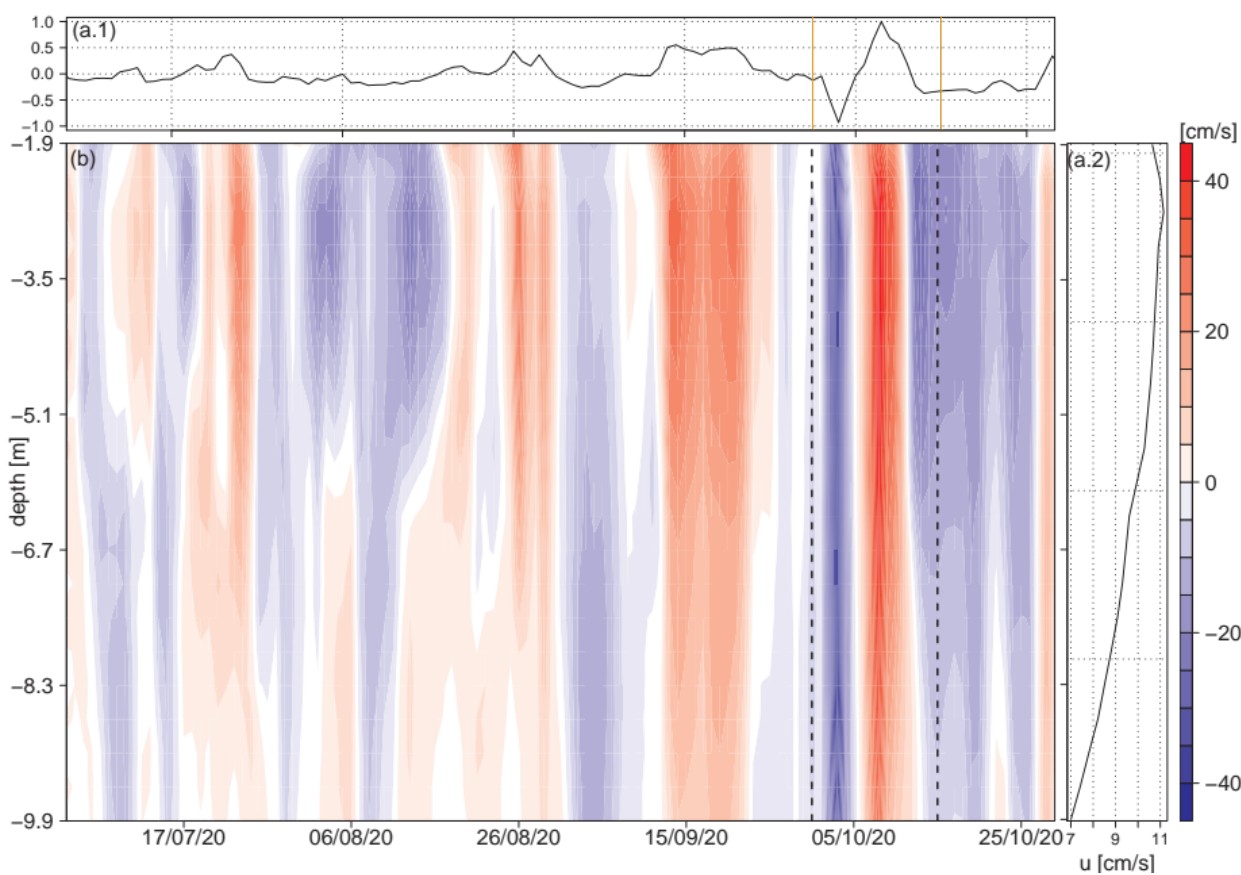

**Figure 4: Mode 1 of the EOF analysis for the ADCP alongshore currents: (a.1) temporal distribution and (a.2) spatial distribution. (b) time-depth distribution of the alongshore current. The dates of the passage of storms Gamma and Delta are highlighted (dashed black and solid orange lines). Red (blue) colors represent a westward (eastward) flow.**


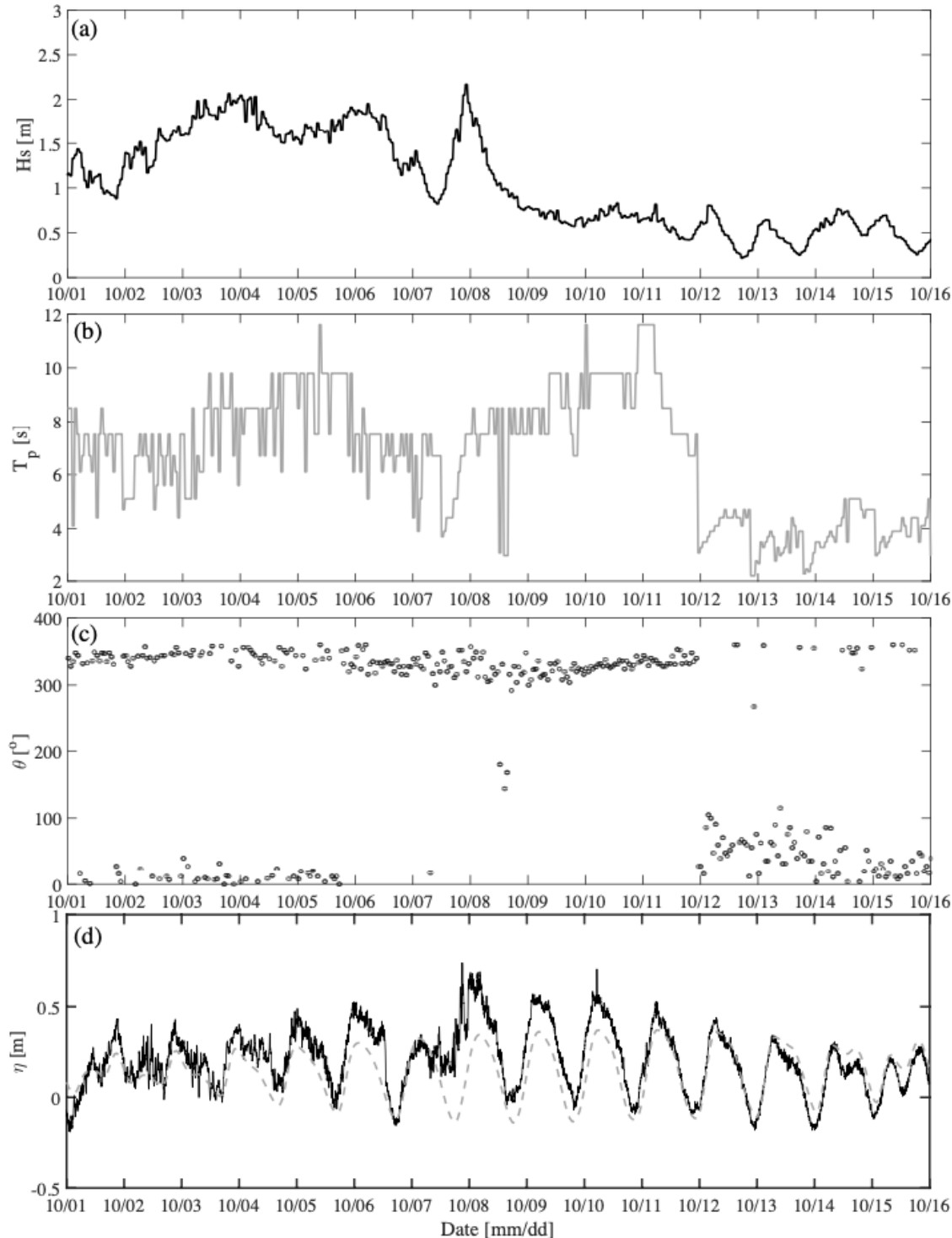

**Figure 5: Time series of (a) significant wave height, (b) peak wave period, (c) mean wave direction measured at the ADCP deployed 10-km offshore, and (d) mean sea level (measured: solid-line; predicted: dashed-line).**

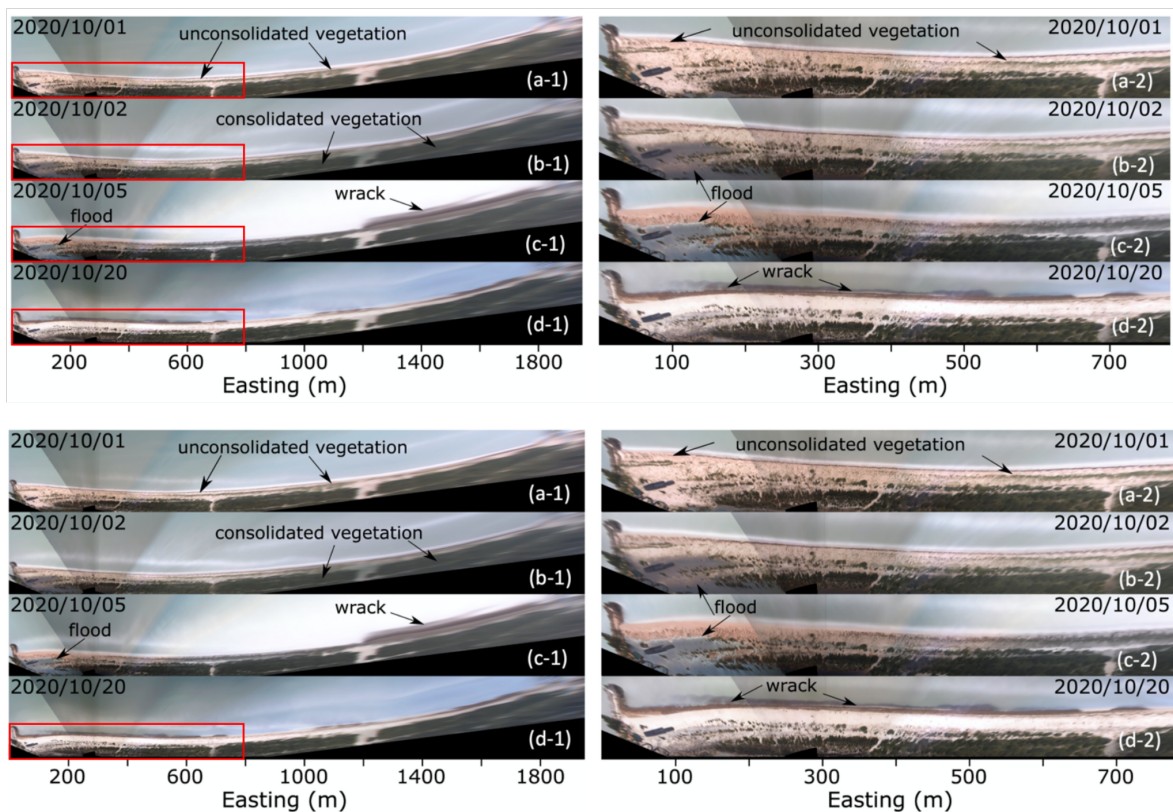

**Figure 6: Rectified images from the video monitoring system showing a (1) 1950 m and (2) 780 m stretch of Sisal beach for (a) 2020/10/01, (b) 2020/10/02, (c) 2020/10/05, and (d) 2020/10/20. Red boxes in (1) represent the area shown in (2). Images taken from: http://tepeu.sisal.unam.mx/**


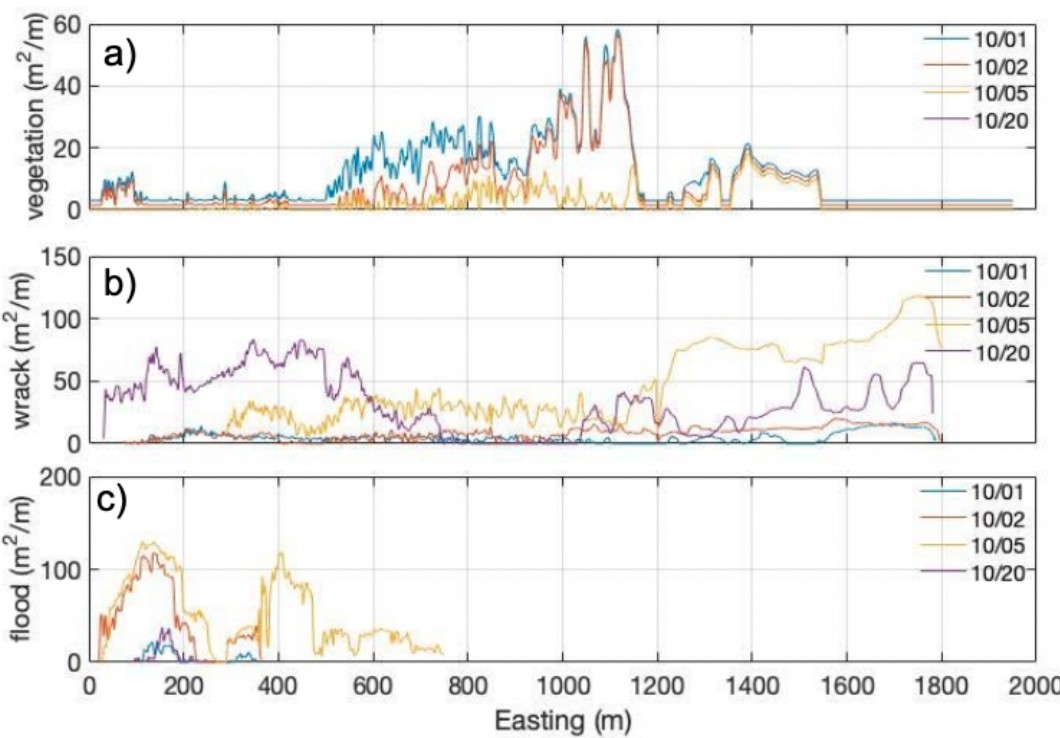

**Figure 7: (a) Wrack, (b) beach berm vegetation, and (c) flood distribution areas along the Sisal beach for 2020/10/01, 2020/10/02, 2020/10/05, and 2020/10/20.**


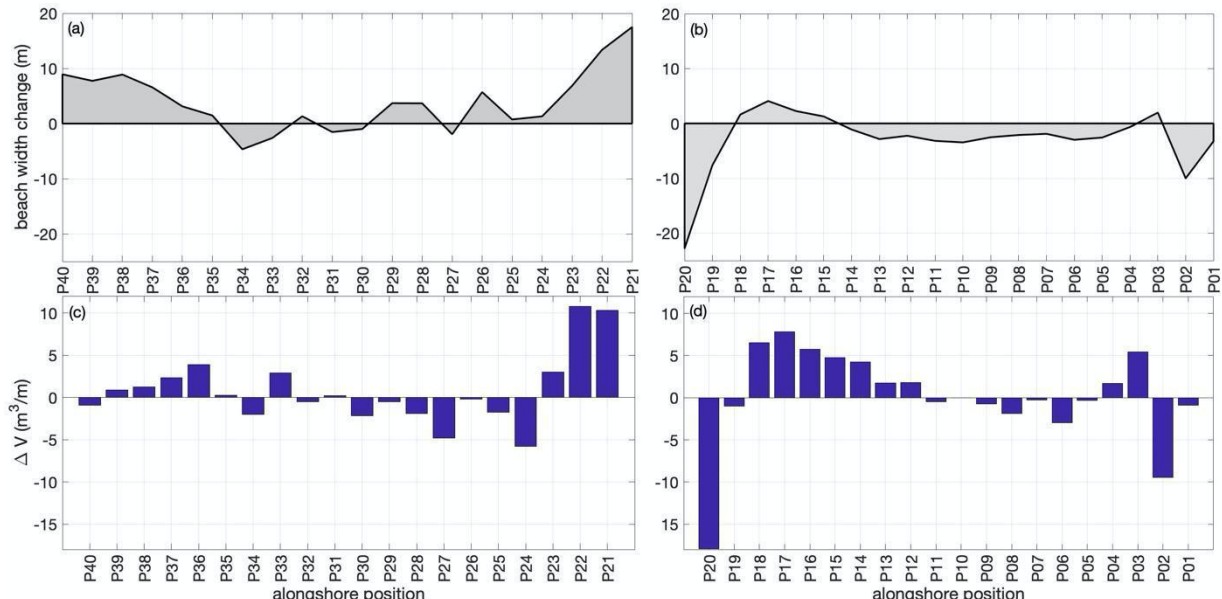

**Figure 8: (a,b)** Shoreline and **(c,d)** subaerial beach volume changes obtained from the DGPS beach profiles east (P01-P20) and west (P21-P40) of Sisal port.


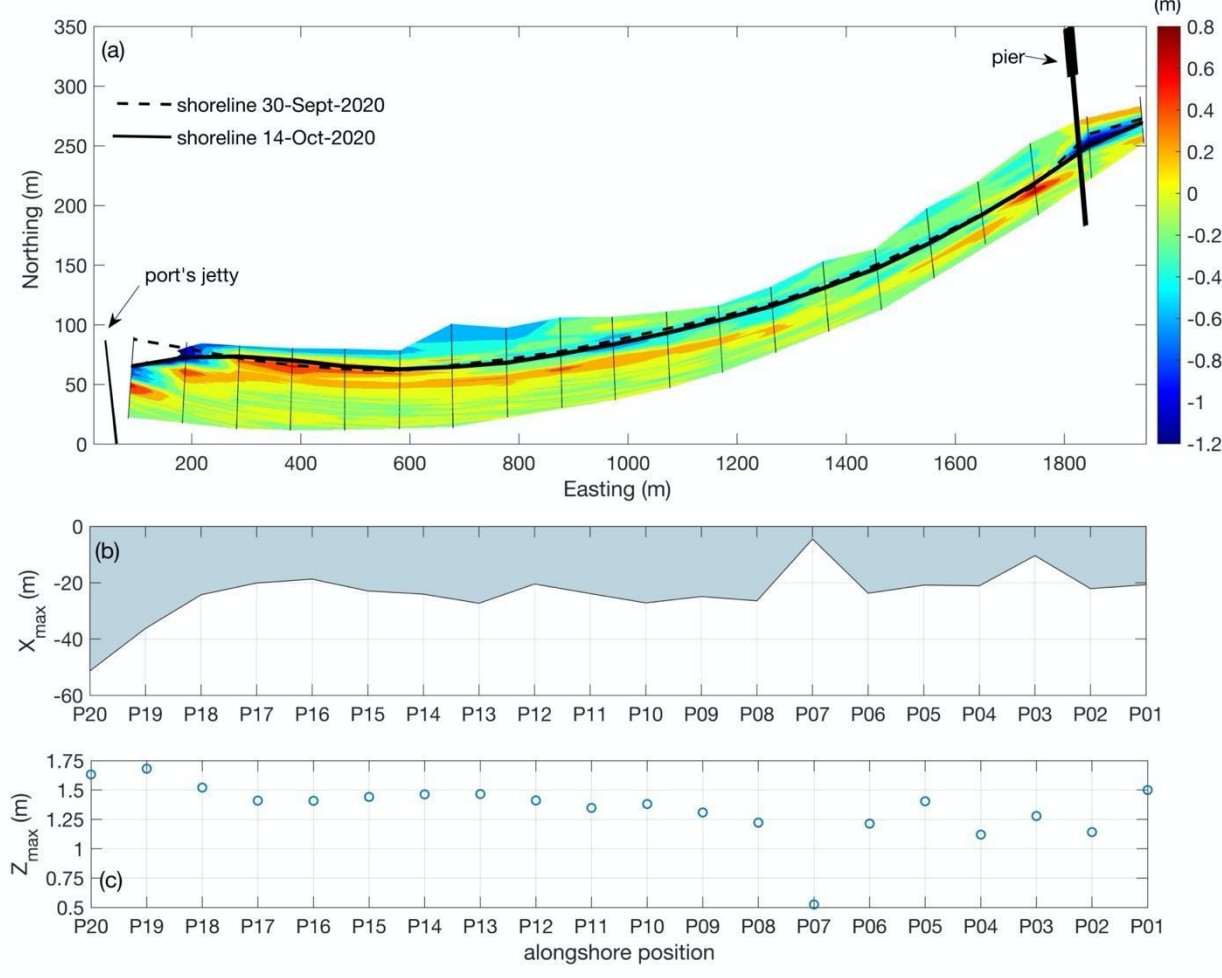

**Figure 9: Coastal flooding derived from beach morphology changes east of Sisal port.**

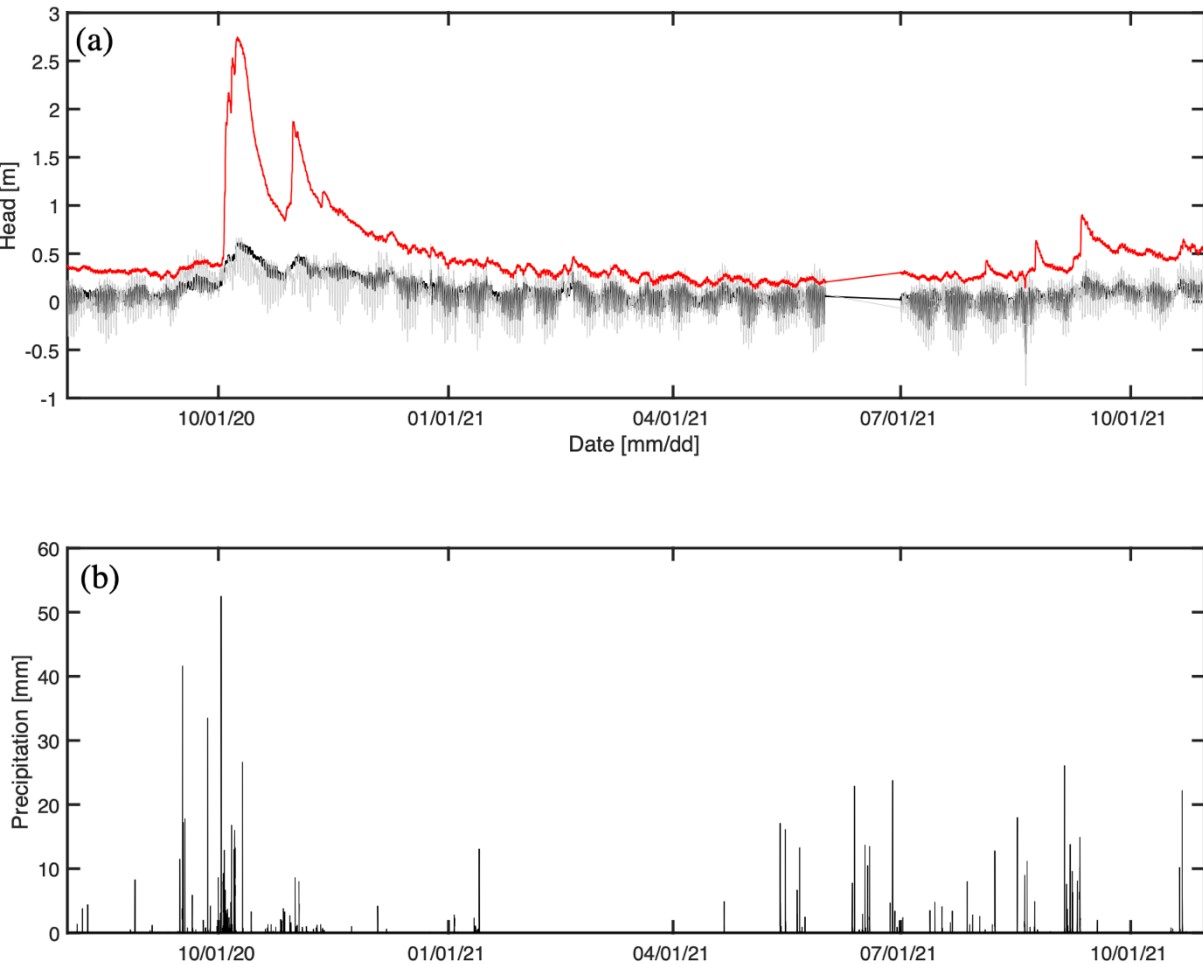


**Figure 10:** Times series of (a) the hydraulic head at three coastal monitoring wells (W4: black solid line; W5: gray solid line; W7a: red line), and (b) precipitation, recorded at the coastal weather station, in Sisal, Mexico. There is missing data due to sensor failure in (a) during June 2021, and (b) from January to April, and from September to October 2021.



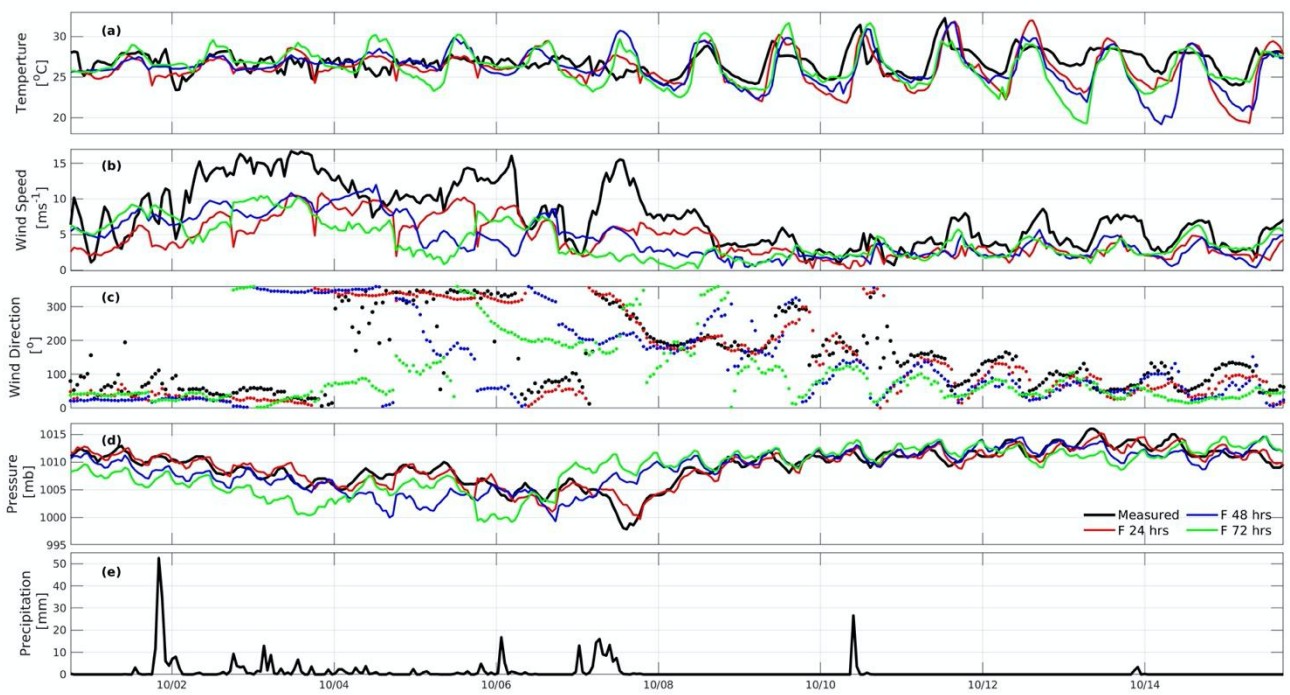

**Figure 11: Measured (black line) and forecast modeled time series of (a) air temperature, (b) wind speed, (c) wind direction, (d) atmospheric pressure, and (e) precipitation. The red line indicates the 24 hr forecast, the blue line the 48 hr, and the green line the 72 hr forecast of the WRF in Sisal, Yucatan from October 1ˢᵗ to 15, 2020.**



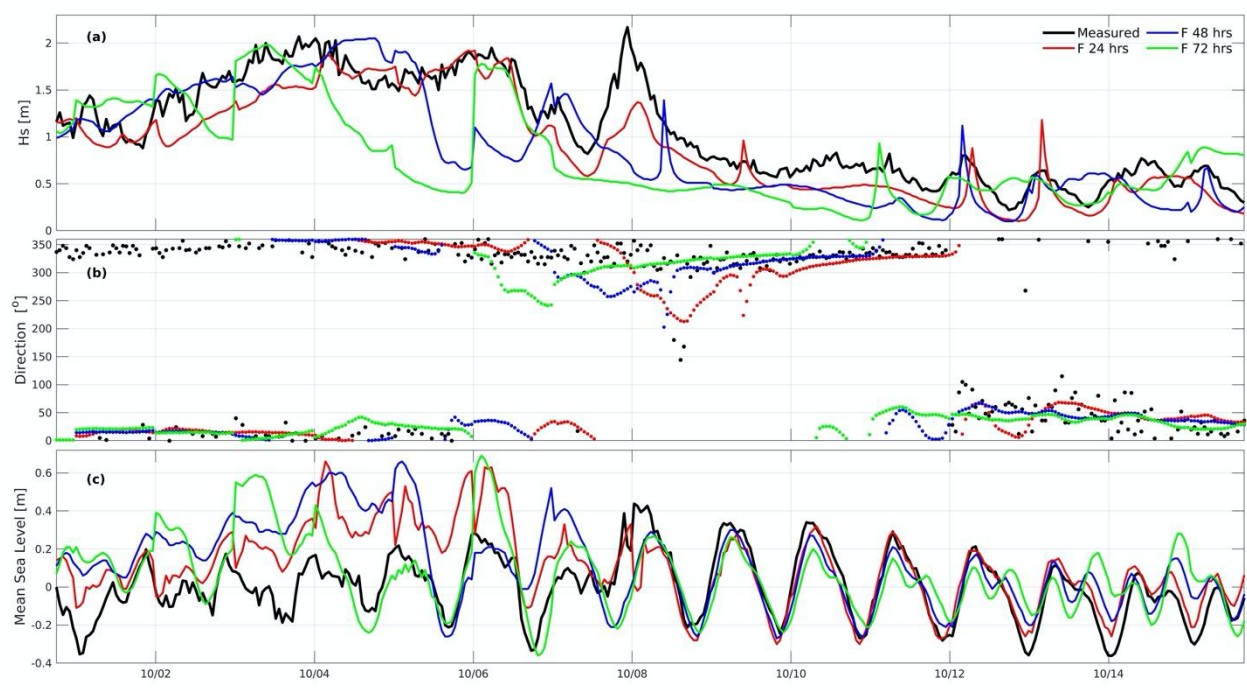


**Figure 12: Measured (black line) and forecast modeled time series of (a) significant wave height, (b) wave direction, and (c) sea level anomaly. The red line indicates the 24 hr forecast, the blue line the 48 hr, and the green line the 72 hr forecast of the WWIII for (a) and (b), and of the ADCIRC model for (c) in Sisal, Yucatan from October 1st to 15, 2020.**


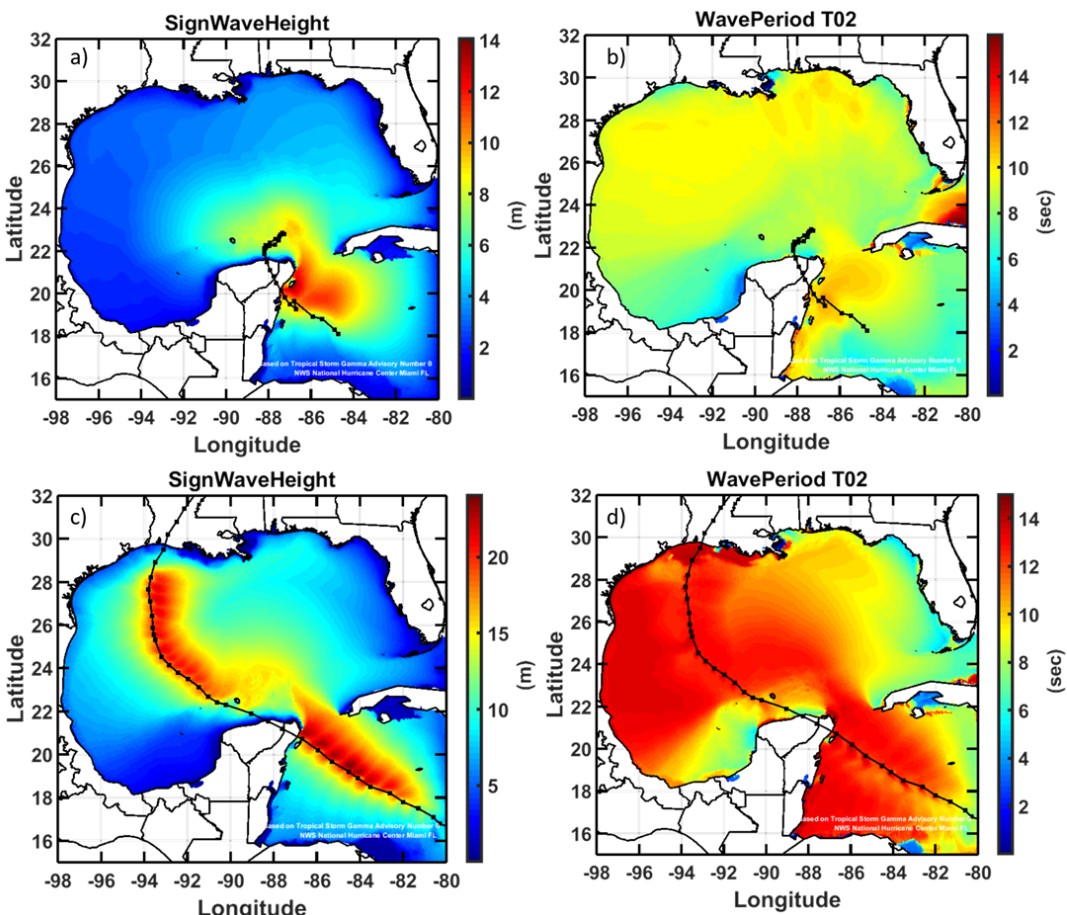

**Figure 13: Envelopes of (a), (c) maximum significant wave height and (b), (d) maximum mean wave period, for both events (a), (b) Gamma and (c), (d) Delta.**


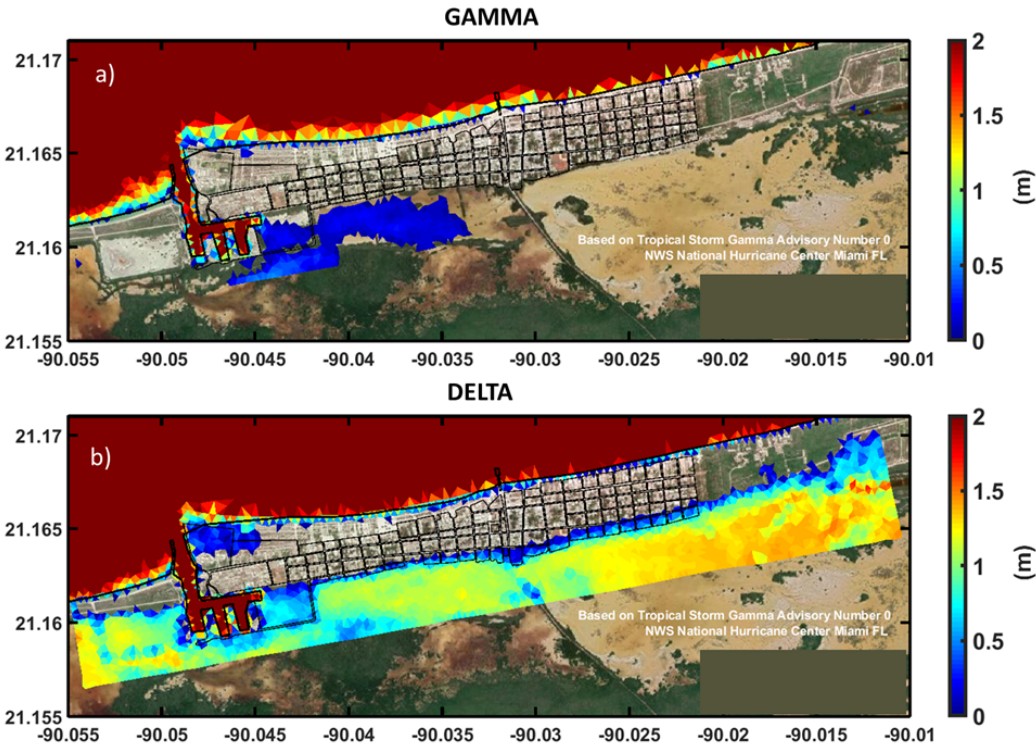

**Figure 14: Maximum envelopes for storm surge generated by events (a) Gamma and (b) Delta at Sisal, Yucatan.**


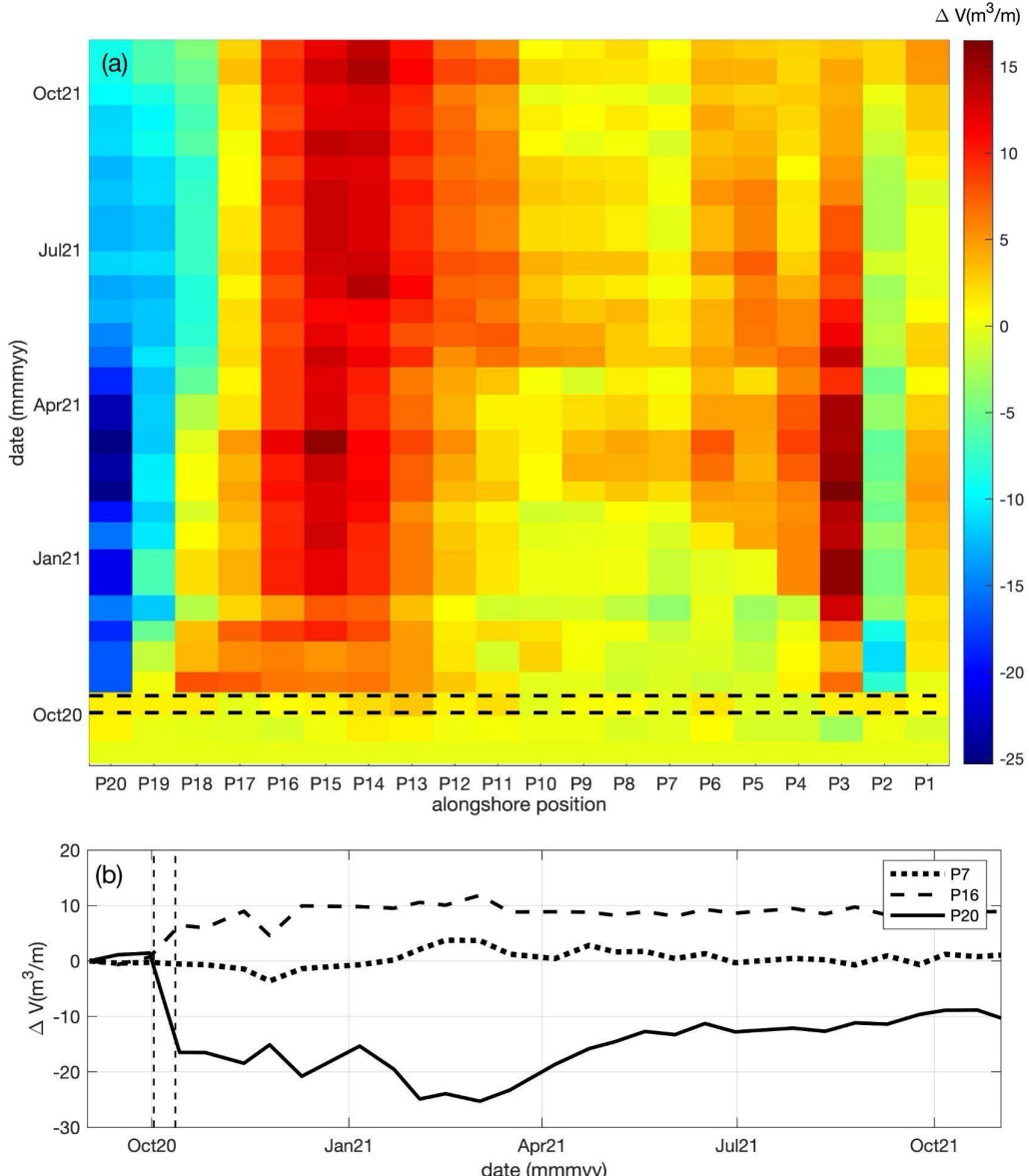

**Figure 15: (a)** Subaerial volume change evolution for beach profiles located east of the port's jetty (storm period for Gamma and Delta: vertical dashed lines) and **(b)** time series of volume change for selected transects with different post-storm behavior (P7-stable, P16-accretion, P-20-erosion/recovery).
