# Peer review of "Hazard assessment and hydrodynamic, morphodynamic, and hydrological response to Hurricanes Gamma and Delta on the northern Yucatan Peninsula"

_Natural Hazards and Earth System Sciences, 2022_

## Referee Comment (RC1)

**Review**

**Introduction**

The problem is clearly defined. It is pertinent and relevant. However, a contextualization of the international context is not made. Although the geographical area where hurricanes occur is limited, it would be necessary to relate (in a very concise way) what approximations or advances have been obtained in such areas.

**Study Area**

For the general contextualization of the study, basic oceanographic information of the study area should be included: regime and range of tides, predominant wave regime, predominant coastal circulation, among other relevant factors.

**Materials and Methods**

What would be the expected accuracy for the DGPS? And in turn, based on this value, what would be the value of the error associated with the estimation of the beach profiles? What is the spatial resolution of the pixels of the video camera system? What was the result of the calibration of this system?

Table 1. Sampling time? Or interval sampling?

LUCL was defined previously?

Why were only the results at 24h, 48h and 72h considered?

**Results**

Pag. 12 -Paragraph 300. The sentence "promoting a sea-level set down (Delta n) of 0.43 m associated with Gamm and a 0.30 m set up during delta" is not completely clear for me considering the scale displayed in the Fig. 2a (36 – 51 cm).

Paragraph 325. Regarding the latent heat (Qe), why are the high values that occur at the end of the time series (10/25/2020) not also mentioned? What factors or phenomena could be associated to these values?

What is the unit of the *x-axis* in Figure 4.a.1? Furthermore, considering Figure 4 and its explanation, the value of 80% for flow towards the west seems excessive. How did you calculate and verify this value?

The editing of Figure 6 is confusing. In subset 1, the area that is presented in detail in subset 2 should be boxed. For example, the jetty is mentioned in the text, but I could not locate it in this figure.

Coastal Aquifer. The increase in the water table is not as noticeable in W5 and W7. Probably due to a figure scale issue. How could the edition of the figure be improved so that this increase can be clearly observed?

Forecast modelling. There is a mention of statistical fit, less bias and significantly high correlation. But the *p-value* is not presented to support these claims. The correlation (C) is the same correlation coefficient (*r*)? If so, wouldn't it be more appropriate to use this denomination?

**Discusión**

The discussion is excessively short and does not cover essential aspects of the manuscript. For example, what would be the main limitations of the approach used? particularly numerical modeling? What effect does the error associated with topographic measurements (i.e. bathymetry, beach profiles, DTM) have on the overall results of the study? If the geographic location of the population with respect to the path of the hurricanes seems to be a crucial factor in the adverse effects, as indicated at the end by including a comparison with another place, why was this approach not included in the experimental design? It was also repeatedly mentioned that anthropic structures marked the transition in morphological changes; then, could it be said that if the structures did not exist, the changes would be more moderate? Would these morphological changes be permanent (long-term) or temporary? Therefore, the discussion must be expanded and restructured, so that the most relevant aspects of the manuscript are addressed.

---

## Author Comment (AC1)

Response to Anonymous Referee #1 comments:
Hazard assessment and hydrodynamic, morphodynamic, and hydrological response to Hurricanes
Gamma and Delta, on the northern Yucatan peninsula (MS No.: nhess-2022-113)
Alec Torres-Freyermuth et al.

Introduction

The problem is clearly defined. It is pertinent and relevant. However, a contextualization of
the international context is not made. Although the geographical area where hurricanes occur
is limited, it would be necessary to relate (in a very concise way) what approximations or
advances have been obtained in such areas.
RESPONSE: We thank the reviewer for pointing out the need to highlight the relevance of
the present work. Significant advances have been achieved in recent decades regarding
hurricane research. While Emanuel (2021) found an increased frequency of tropical cyclones
in the North Atlantic, there is no clear trend in the increase of tropical cyclone frequency due
to climate change. Nevertheless, most studies find an increased proportion of the most
extreme events (i.e., categories 4 and 5 on the Saffir Simpson scale) in the context of climate
change (Knutson et al., 2020), increasing the associated hazards by the second half of the
century. Furthermore, a poleward migration of the location of the maximum lifetime intensity
of tropical cyclones has been found (Kossin, 2014), as well as an increase in rapid
intensification (Bhatia et al., 2019; Emanuel, 2017),  increasing the hazards from tropical
cyclones in higher latitudes and hence representing a challenge for emergency management.
Regarding the impacts of hurricanes in coastal areas, recent studies have pointed out the need
to investigate storm impacts from an interdisciplinary point of view (Camelo and Tamayo,
2021). Considering the importance of numerical modelling for the development of early
warning systems and emergency management plans, an improvement in the understanding
of the feedback between terrestrial-atmospheric-oceanic processes needs to be incorporated
into the numerical models to estimate their impacts in coastal areas. This is a challenging
task for a low-lying (karstic) coast where the groundwater aquifer discharges towards the
coast. In summary, progress in the understanding of climate change effects on the
geographical distribution of hurricanes and the importance of conducting interdisciplinary
reaseach to improve predictive tools will be highlight in the revised manuscript.

Study Area

For the general contextualization of the study, basic oceanographic information of the study
area should be included: regime and range of tides, predominant wave regime, predominant
coastal circulation, among other relevant factors.
RESPONSE: We agree that this section should be improved with a better description of the
environmental characteristics. The mean sea level presents a seasonal variability ascribed to
alongshelf currents on the western Gulf of Mexico and low-frequency atmospheric pressure
variability (Zavala-Hidalgo et al., 2003). The tidal regime in the study area is diurnal micro-
tidal, with a spring tidal range of 0.75 m (Valle-Levinson et al., 2011). The study area is
characterized by intense sea breeze events (Figueroa et al., 2014) and winter storms (fall-
winter) associated with Central American Cold Surge Events (Medina-Gómez & Herrera-

Silveira, 2009; Kurzcyn et al., 2020). Typical winter storm wave conditions reach Hs>2 m and Tp>8s from the NNW. Sea breezes drive low-energy high-angle short-period waves (Hs<1 m, Tp=3.5 s, NE) that drive a persistent westward alongshore current (Torres-Freyermuth et al., 2017). Across the Yucatán shelf, winds and mesoscale circulation are responsible for driving the currents mainly toward the west (Enriquez et al., 2010; Torres-Freyermuth et al., 2017).

Materials and Methods
What would be the expected accuracy for the DGPS? And in turn, based on this value, what would be the value of the error associated with the estimation of the beach profiles?
RESPONSE: The DGPS employed Real Time Kinematics (RTK) with a horizontal and vertical accuracy of 0.010 and 0.020 m, respectively, as reported by the manufacturer. Additionally, we carried out a test to assess the accuracy of the method employed by comparing elevation measurements taken along a straight line on a parking lot with the GPS rover fixed on a pole with a bubble level vs carrying the GPS rover in a backpack by two different users (see Figure 1). Maximum differences in the vertical measurements were less than 0.04 m. Therefore, we expect that the maximum error associated with estimating the beach profiles' elevation is of that magnitude.

[Figure]

Figure 1. Elevation measurements using a GPS rover fixed on a pole with a bubble level (control) vs. measurements carrying the rover in a backpack by two different users.

What is the spatial resolution of the pixels of the video camera system? What was the result of the calibration of this system?
RESPONSE: The spatial resolution of the different components of this camera system has been previously described by Arriaga et al. (2022). Given that here we are interested mainly in coverage, this result has to be translated into areas. For example, along the shoreline near the cameras (300 m easting), a pixel translates to an area of 0.01 m². Around 1400 m easting, a pixel covers an approximate size of 2.5 m². Finally, at the farthest point, near the pier, the resolution is in the order of 10 m². The calibration performed following Simarro et al. 2017, particularly the simplified mathematical model referred to as M2 in Simarro et al. 2020, resulted in an average reprojection error of 0.7 pixels.

Table 1. Sampling time? Or interval sampling?

RESPONSE: The Table has been revised accordingly by replacing "Sampling time" with "Sampling interval."

LUCL was defined previously?
RESPONSE: Thank you for the observation. LULC means WRF Land Use and Land Cover (LULC) soil category maps. The definition will be included in the revised manuscript.

Why were only the results at 24h, 48h, and 72h considered?
RESPONSE: It is known that a tropical cyclone's track and intensity forecast degrades rapidly, so it is not reliable beyond 72 hours. Despite the performance of the forecasts being acceptable for this particular case (see Table 1), it was not considered for discussion. The National Hurricane Center Forecast Verification Report (https://www.nhc.noaa.gov/verification/pdfs/Verification_2020.pdf), 2020 Hurricane Season, shows the error in forecasts can vary significantly from year to year due to the interannual variability of the atmosphere. According to the NHC 2020 forecasts (see Figure 2), the average error of a tropical cyclones' intensity is observed as a function of the forecast time.

Table. Correlation Coefficients of 24 to 120 hrs forecasts.

| Variables | Correlation Coefficient | | | | |
| | 24 hrs | 48 hrs | 72 hrs | 96 hrs | 120 hrs |
| --- | --- | --- | --- | --- | --- |
| Temperature | 0.58 | 0.40 | 0.39 | 0.43 | 0.44 |
| Wind Speed | 0.81 | 0.69 | 0.56 | 0.50 | 0.28 |
| Wind Direction | 0.66 | 0.50 | 0.24 | 0.19 | 0.04 |
| Atmospheric Pressure | 0.93 | 0.70 | 0.56 | 0.44 | 0.38 |
| Significant Height  (Hs) | 0.94 | 0.77 | 0.64 | 0.51 | 0.32 |
| Hs Direction | 0.67 | 0.63 | 0.58 | 0.40 | 0.14 |
| Sea Level | 0.64 | 0.46 | 0.38 | 0.20 | 0.01 |

[Figure]

[Figure]

Figure 2. (Left panel) Cumulative distribution of five-year official Atlantic basin tropical cyclone track forecast errors (https://www.nhc.noaa.gov/verification/verify4.shtml). (Right panel) 2020 NHC official intensity errors for all tropical cyclones at 24, 72, and 120 h (https://www.nhc.noaa.gov/verification/pdfs/Verification_2020.pdf).

Results

Pag. 12 -Paragraph 300. The sentence "promoting a sea-level set down (Delta n) of 0.43 m associated with Gamma and a 0.30 m set up during delta" is not completely clear for me considering the scale displayed in the Fig. 2a (36 – 51 cm).

RESPONSE: We apologize for the lack of clarity in the figure/text in this section. The scale displayed is between -36 to +51 cm (86 cm total). The revised text should read:

*"promoting a sea-level set up (Delta n) of 9 cm associated with Gamma and a 30 cm set up during Delta."* This information will be included in the revised manuscript.

Paragraph 325. Regarding the latent heat (Qe), why are the high values that occur at the end of the time series (10/25/2020) not also mentioned? What factors or phenomena could be associated to these values?

RESPONSE: Latent Heat is the flux of energy carried by evaporated water. October is part of the rainy season for the region, where moisture condenses into rain drops, forming clouds and taking latent heat from the ocean's surface. Therefore, we believe that cloud formation could be associated with higher values at the end of the time series.

What is the unit of the x-axis in Figure 4.a.1? Furthermore, considering Figure 4 and its explanation, the value of 80% for flow towards the west seems excessive. How did you calculate and verify this value?

RESPONSE: The x-axis in Figure 4.a.1 is the same as in Figure 4.b (i.e., dates). The period shown in Figure 4 may be misleading for westward flow. The currents of the Yucatan Peninsula are predominantly westward. The value of 80% was calculated considering the three years time series (which included the hurricane events), during which a zonal

component showed a negative flow (ie. towards the west). This clarification will be included in the revised manuscript.

The editing of Figure 6 is confusing. In subset 1, the area that is presented in detail in subset 2 should be boxed. For example, the jetty is mentioned in the text, but I could not locate it in this figure.

RESPONSE: The figure has been revised following the reviewer's suggestions. Notice that the port's jetty is located on the left side of both subsets as a reference.

[Figure]

Figure 3. Rectified images from the video monitoring system showing a (1) 1950 m and (2) 780 m stretch of Sisal beach for (a) 2020/10/01, (b) 2020/10/02, (c) 2020/10/05, and (d) 2020/10/20. Images taken from: http://tepeu.sisal.unam.mx/

Coastal Aquifer. The increase in the water table is not as noticeable in W5 and W7. Probably due to a figure scale issue. How could the edition of the figure be improved so that this increase can be clearly observed?

RESPONSE: The figure has been revised accordingly to highlight the increase in the water table at monitoring wells W5 and W7. We increased the period (08/01/2020-10/30/2020) so that the impact in the aquifer could be best observed (see Figure 4 below). As the reviewer pointed out, the changes in W7 and W5 are not as significant as in W4, where the maximum

increase is about 0.5 m. However, there was also a decrease in the amplitude of the measured tides at the wells caused by the increase in aquifer discharge.

[Figure]

Figure 4. Times series of (a) the hydraulic head at three coastal monitoring wells (W4: solid black line; W5: black dotted line; W7a: solid gray line), and (b) precipitation at the RUOA meteorological station in Sisal, Mexico.

Therefore, the text from Section 4.2.3 will be revised as follows:

*"Monitoring wells W4, W5, and W7a located 20 km, 5 km, and 200 m from the coast, respectively, provide information on the oceanic and terrestrial forcing on the coastal aquifer. Figure 10 shows the relative levels of the hydraulic head at each well and the precipitation at Sisal. Wells W7a, and W5 show the diurnal tidal modulation on the hydraulic head (Figure 10a). All wells show an increase in the water table owing to the recharge*

*following the storms. This is more evident at the well located 20 km from Sisal (W4), which shows an increase of more than 2 m following the passage of Gamma and Delta, reaching a maximum level on October 7. It is worth noting that coastal wells W5 and W7a do not show such an increase in the water table due to confined aquifer conditions that do not allow rapid infiltration of the precipitation (Figure 10b); however, the storm effects can be seen in the change in amplitude (decrease) of the tide caused by the increase in aquifer discharge. The southern limit of such confinement is not well known, hence back-barrier flooding might occur when the water table exceeds the confinement level south of the aquifer, preventing the hydraulic head in well W7a and W5 from increasing further (Perry, 1989; Pino 2011)."*

Forecast modelling. There is a mention of statistical fit, less bias and significantly high correlation. But the p-value is not presented to support these claims. The correlation (C) is the same correlation coefficient (r)? If so, wouldn't it be more appropriate to use this denomination?

RESPONSE: Thank you for your observations. The manuscript has been revised, replacing "correlation" by "correlation coefficient". Regarding the statistical parameters, a new Table will be included in the manuscript to illustrate our results (Table1).

Table 1. Statistical parameters at the 5% significance level. df: degree of freedom; sd: standard deviation; ci: confidence interval.

| Variables | t-test | df | sd | p-value | ci | |
|---|---|---|---|---|---|---|
| Temperature | 13.17 | 4 | 0.08 | 0.0001 | 0.35 | 0.54 |
| Wind Speed | 6.29 | 4 | 0.20 | 0.003 | 0.32 | 0.82 |
| Wind Direction | 2.96 | 4 | 0.25 | 0.042 | 0.02 | 0.64 |
| Atmospheric Pressure | 6.14 | 4 | 0.22 | 0.004 | 0.33 | 0.88 |
| Significant Height (Hs) | 6.01 | 4 | 0.24 | 0.004 | 0.34 | 0.93 |
| Hs Direction | 4.99 | 4 | 0.22 | 0.008 | 0.22 | 0.75 |
| Sea Level | 3.12 | 4 | 0.24 | 0.036 | 0.04 | 0.64 |

Discussion
The discussion is excessively short and does not cover essential aspects of the manuscript.
RESPONSE: We thank the reviewer for pointing out this issue. All the reviewer's comments are considered for the revised version of the manuscript. See responses below.

For example, what would be the main limitations of the approach used? particularly numerical modeling?

RESPONSE: The numerical modelling presents limitations in forcing and incorporating terrestrial and atmospheric processes. The spatial and temporal resolution for the wind field associated with the storm passage is not high enough to capture the hydrodynamic response near the coast for such events that move/travel offshore. On the other hand, field observations suggest that the flooding on the barrier island's lee side was associated with the high precipitation and the increase in the water table. These two aspects were not incorporated into the numerical models. We believe that water level measurements in the wetlands would provide greater insights into the importance of precipitation in the flooding of the barrier island.

What effect does the error associated with topographic measurements (i.e. bathymetry, beach profiles, DTM) have on the overall results of the study?

RESPONSE: The uncertainty in the topography and bathymetry is important for the model implementation since it controls wave transformation and coastal flooding. However, we found that our DGPS measurement errors are significantly smaller than the observed changes in the beach and hence do not affect the conclusions reached in the present study.

If the geographic location of the population with respect to the path of the hurricanes seems to be a crucial factor in the adverse effects, as indicated at the end by including a comparison with another place, why was this approach not included in the experimental design?

RESPONSE: We agree that the geographical location can play an important role in the observed impact. In the present work, we focus on obtaining high-resolution measurements that can help to calibrate numerical models that can be implemented in other areas. However, we believe that comparing field observations with other places along the northern Yucatan coast warrants future studies.

It was also repeatedly mentioned that anthropic structures marked the transition in morphological changes; then, could it be said that if the structures did not exist, the changes would be more moderate?

RESPONSE: We believe the impact of the structures is quite important in the study area as confirmed by aerial images and in situ observations over the past few years. Observed morphological changes are significantly smaller at beach profiles located away from the structures. Therefore, it is likely that, as the reviewer points out, if the structures did not exist, extreme changes would be more moderate.

Would these morphological changes be permanent (long-term) or temporary?

RESPONSE: We agree that this is an important aspect that can be addressed with our beach profile measurements. To investigate this, we analyzed the beach morphology changes occurring during the following year after the storm passage (Figure 5). It was found that some morphological features remained, such as the increase in the beach elevation in some areas, whereas other changes smoothed out during the winter storm season. Moreover, the beach evolution presented significant alongshore differences including a net volume gain (P16), fast recovery (P7), and a slow recovery (P20) after the storm passage (see Figure 5b).

[Figure]

Figure 5. (a) Volume change evolution for each beach profile indicating the storm period for Gamma and Delta (dashed lines) and (b) time series of volume change for selected transects with different post-storm behavior (P7-stable, P16-accretion, P-20-erosion/recovery).

Therefore, the discussion must be expanded and restructured, so that the most relevant aspects of the manuscript are addressed.

RESPONSE: Following the reviewer's comments, the Discussion section has been re-structured in the revised manuscript to include the aforementioned points raised by the reviewer.

REFERENCES:

Bhatia, K. T., Vecchi, G. A., Knutson, T. R., Murakami, H., Kossin, J., Dixon, K. W., & Whitlock, C. E. (2019). Recent increases in tropical cyclone intensification rates. Nature Communications, 10(1), 1–9. https://doi.org/10.1038/s41467-019-08471-z

Emanuel, K. (2017). Will global warming make hurricane forecasting more difficult? Bulletin of the American Meteorological Society, 98(3), 495–501. https://doi.org/10.1175/BAMS-D-16-0134.1

Emanuel, K. (2021). Atlantic tropical cyclones downscaled from climate reanalyses show increasing activity over past 150 years. Nature Communications, (1), 1–8. https://doi.org/10.1038/s41467-021-27364-8

Kossin, J. P., Emanuel, K. A., Vecchi, G. A. (2014). The poleward migration of the location of tropical cyclone maximum intensity. Nature, 509(7500), 349–352. https://doi.org/10.1038/nature13278

Kossin, J. P., Emanuel, K. A., Vecchi, G. A. (2014). The poleward migration of the location of tropical cyclone maximum intensity. Nature, 509(7500), 349–352. https://doi.org/10.1038/nature13278

---

## Author Response (AR1)

Response to Anonymous Referee #1 comments:
Hazard assessment and hydrodynamic, morphodynamic, and hydrological response to Hurricanes Gamma and Delta, on the northern Yucatan peninsula (MS No.: nhess-2022-113)
Alec Torres-Freyermuth et al.

Introduction

The problem is clearly defined. It is pertinent and relevant. However, a contextualization of the international context is not made. Although the geographical area where hurricanes occur is limited, it would be necessary to relate (in a very concise way) what approximations or advances have been obtained in such areas.
RESPONSE: We thank the reviewer for pointing out the need to highlight the relevance of the present work. Significant advances have been achieved in recent decades regarding hurricane research. The following paragraphs have been included in the revised manuscript:

Lines 39-46:

*"Significant advances have been achieved in recent decades regarding hurricane research. While Emanuel (2021) found an increased frequency of tropical cyclones in the North Atlantic, there is no clear trend in the increase of tropical cyclone frequency due to climate change. Nevertheless, most studies find an increased proportion of the most extreme events (i.e., categories 4 and 5 on the Saffir Simpson scale) in the context of climate change (Knutson et al., 2020), increasing the associated hazards by the second half of the century. Furthermore, a poleward migration of the location of the maximum lifetime intensity of tropical cyclones has been found (Kossin, 2014), as well as an increase in rapid intensification (Bhatia et al., 2019; Emanuel, 2017), increasing the hazards from tropical cyclones in higher latitudes and hence representing a challenge for emergency management."*

Lines 53-57:

*"Moreover, recent studies have pointed out the need to investigate storm impacts from an interdisciplinary point of view (Camelo and Tamayo, 2021). Considering the importance of numerical modelling for developing early warning systems and emergency management plans, an improvement in the understanding of the feedback between terrestrial-atmospheric-oceanic processes needs to be incorporated into numerical models to estimate their impacts in coastal areas. This is a challenging task for a low-lying (karstic) coast where the groundwater aquifer discharges towards the shore."*

In summary, progress in the understanding of climate change effects on the geographical distribution of hurricanes, and the importance of conducting interdisciplinary reaseach to improve predictive tools are now highlight in the revised manuscript.

Study Area

For the general contextualization of the study, basic oceanographic information of the study area should be included: regime and range of tides, predominant wave regime, predominant coastal circulation, among other relevant factors.
RESPONSE: We agree that this section should be improved with a better description of the environmental characteristics. The following information has bee included in the revised manuscript (Lines 87-94):

*"The study area is characterized by intense sea breeze events (Figueroa et al., 2014) and winter storms (fall-winter) associated with Central American Cold Surge Events (Medina-Gómez & Herrera-Silveira, 2009; Kurzcyn et al., 2020). Typical winter storm wave conditions reach Hs>2 m and Tp>8 s from the NNW. Sea breezes drive low-energy high-angle short-period waves (Hs<1 m, Tp=3.5 s, NE) that drive a persistent westward alongshore current (Torres-Freyermuth et al., 2017). Across the Yucatan shelf, winds and mesoscale circulation drive the currents mainly toward the west (Enriquez et al., 2010; Torres-Freyermuth et al., 2017). The mean sea level in this area presents a seasonal variability ascribed to alongshelf currents on the western Gulf of Mexico and low-frequency atmospheric pressure variability (Zavala-Hidalgo et al., 2003). The tidal regime in the study area is diurnal micro-tidal, with a spring tidal range of 0.75 m (Valle-Levinson et al., 2011)."*

Materials and Methods
What would be the expected accuracy for the DGPS? And in turn, based on this value, what would be the value of the error associated with the estimation of the beach profiles?
RESPONSE: The DGPS-RTK measurements have a horizontal and vertical accuracy of 0.010 and 0.020 m, respectively, as reported by the manufacturer. Additionally, a test was carried out to assess the method`s accuracy by comparing elevation measurements taken along a straight line on a parking lot with the GPS rover fixed on a pole with a bubble level against carrying the GPS rover in a backpack by two different users. Maximum differences in the vertical measurements were less than 0.04 m. Therefore, we expect that the maximum error associated with estimating the beach profiles' elevation is of that magnitude. This information has been included in the manuscript (Lines 125-130).

[Figure]

**Figure 1: Elevation measurements using a GPS rover fixed on a pole with a bubble level (control) vs. measurements carrying the rover in a backpack by two different users.**

What is the spatial resolution of the pixels of the video camera system? What was the result of the calibration of this system?

RESPONSE: The spatial resolution of the different components of this camera system has been previously described by Arriaga et al. (2022). Given that here we are interested mainly in coverage, this result has to be translated into areas. For example, along the shoreline near the cameras (300 m easting), a pixel translates to an area of 0.01 $m^2$, whereas 1400 m easting a pixel covers an approximate size of 2.5 $m^2$. Finally, at the farthest point, near the pier, the resolution is in the order of 10 $m^2$. The calibration performed following Simarro et al. (2017), particularly the simplified mathematical model referred to as M2 in Simarro et al. (2020), resulted in an average reprojection error of 0.7 pixels. This information is now included in Lines 134-140.

Table 1. Sampling time? Or interval sampling?

RESPONSE: The Table has been revised accordingly by replacing "Sampling time" with "Sampling interval."

LUCL was defined previously?

RESPONSE: Thank you for the observation. LULC means Land Use and Land Cover soil category maps. The definition has been included in the revised manuscript (line 272).

Why were only the results at 24h, 48h, and 72h considered?

RESPONSE: It is known that a tropical cyclone's track and intensity forecast degrades rapidly, so it is not reliable beyond 72 hours. Despite the performance of the forecasts being acceptable for this particular case (see Table 1), it was not considered for discussion. The National Hurricane Center Forecast Verification Report (https://www.nhc.noaa.gov/verification/pdfs/Verification_2020.pdf), 2020 Hurricane Season, shows the error in forecasts can vary significantly from year to year due to the interannual variability of the atmosphere. According to the NHC 2020 forecasts (see Figure 2), the average error of a tropical cyclones' intensity is observed as a function of the forecast time.

Table. Correlation Coefficients of 24 to 120 hrs forecasts.

| | Correlation Coefficient | | | | |
|---|---|---|---|---|---|
| Variables | 24 hrs | 48 hrs | 72 hrs | 96 hrs | 120 hrs |
| Temperature | 0.58 | 0.40 | 0.39 | 0.43 | 0.44 |
| Wind Speed | 0.81 | 0.69 | 0.56 | 0.50 | 0.28 |

| | | | | | |
|---|---|---|---|---|---|
| Wind Direction | 0.66 | 0.50 | 0.24 | 0.19 | 0.04 |
| Atmospheric Pressure | 0.93 | 0.70 | 0.56 | 0.44 | 0.38 |
| Significant Height  (Hs) | 0.94 | 0.77 | 0.64 | 0.51 | 0.32 |
| Hs Direction | 0.67 | 0.63 | 0.58 | 0.40 | 0.14 |
| Sea Level | 0.64 | 0.46 | 0.38 | 0.20 | 0.01 |

[Figure]

**Figure 2: (Left panel) Cumulative distribution of five-year official Atlantic basin tropical cyclone track forecast errors (https://www.nhc.noaa.gov/verification/verify4.shtml). (Right panel) 2020 NHC official intensity errors for all tropical cyclones at 24, 72, and 120 h (https://www.nhc.noaa.gov/verification/pdfs/Verification_2020.pdf).**

Results

Pag. 12 -Paragraph 300. The sentence "promoting a sea-level set down (Delta n) of 0.43 m associated with Gamma and a 0.30 m set up during delta" is not completely clear for me considering the scale displayed in the Fig. 2a (36 – 51 cm).

RESPONSE: We apologize for the lack of clarity in the figure/text in this section. The figure (Figure 3a) and text (Lines ) have been revised (Lines 351-353):

*The winds accelerated to reach 14.4 m s-1 (52 km h-1), blowing from the north and promoting a sea-level set up (Δη) of 9 cm associated with Gamma and a 30 cm set up during Delta. Ocean currents increased to reach the maximum value registered in the time series since 2018 (59 cm s-1) during Gamma (Figure 3b).*

[Figure]

Figure 3: Time series of (a) wind velocity and sea surface height (in blue), (b) current velocity and significant wave height (in red), (c) Ekman currents and Ekman layer depth (in red), (d) sea bottom temperature and air temperature (in red), (e) seawater density and precipitation (in red), and (f) sensible and latent heat (in red), estimated from the meteorological station, the moored ADCP and the satellite data. The dates of the passage of storms Gamma and Delta are highlighted.

Paragraph 325. Regarding the latent heat (Qe), why are the high values that occur at the end of the time series (10/25/2020) not also mentioned? What factors or phenomena could be associated to these values?

RESPONSE: Latent Heat is the flux of energy carried by evaporated water. October is part of the rainy season for the region, where moisture condenses into rain drops, forming clouds and taking latent heat from the ocean's surface. Therefore, we believe that cloud formation could be associated with higher values at the end of the time series (see Lines 376-379).

What is the unit of the x-axis in Figure 4.a.1? Furthermore, considering Figure 4 and its explanation, the value of 80% for flow towards the west seems excessive. How did you calculate and verify this value?

RESPONSE: The x-axis in Figure 4.a.1 is the same as in Figure 4.b (i.e., dates). The period shown in Figure 4 may be misleading for westward flow. The currents of the Yucatan Peninsula are predominantly westward. The value of 80% was calculated considering the three years time series (which included the hurricane events), during which a zonal component showed a negative flow (ie. towards the west). This clarification will be included in the revised manuscript.

The editing of Figure 6 is confusing. In subset 1, the area that is presented in detail in subset 2 should be boxed. For example, the jetty is mentioned in the text, but I could not locate it in this figure.

RESPONSE: The figure has been revised following the reviewer's suggestions (see Figure 4 below and Figure 6 in the revised manuscript). Notice that the port's jetty is located on the left side of both subsets as a reference.

[Figure]

**Figure 4: Rectified images from the video monitoring system showing a (1) 1950 m and (2) 780 m stretch of Sisal beach for (a) 2020/10/01, (b) 2020/10/02, (c) 2020/10/05, and (d) 2020/10/20. Images taken from: http://tepeu.sisal.unam.mx/**

Coastal Aquifer. The increase in the water table is not as noticeable in W5 and W7. Probably due to a figure scale issue. How could the edition of the figure be improved so that this increase can be clearly observed?

RESPONSE: The figure has been revised accordingly to highlight the increase in the water table at monitoring wells W5 and W7. We increased the period (08/01/2020-10/30/2020) so that the impact in the aquifer could be best observed (see Figure 5 below and Figure 10 in the revised manuscript). As the reviewer pointed out, the changes in W7 and W5 are not as significant as in W4, where the maximum increase is about 0.5 m. However, there was also a decrease in the amplitude of the measured tides at the wells caused by the increase in aquifer discharge.

[Figure]

**Figure 5: Times series of (a) the hydraulic head at three coastal monitoring wells (W4: black solid line; W5: gray solid line; W7a: red line), and (b) precipitation, recorded at the coastal weather station, in Sisal, Mexico. There is missing data due to sensor failure in (a) during June 2021, and (b) from January to April, and from September to October 2021.**

Therefore, the text from Section 4.2.3 will be revised as follows (Lines 452-462):

*"Monitoring wells W4, W5, and W7a located 20 km, 5 km, and 200 m from the coast, respectively, provide information on the oceanic and terrestrial forcing on the coastal aquifer. Figure 10 shows the relative levels of the hydraulic head at each well and the precipitation at Sisal. Wells W7a, and W5 show the diurnal tidal modulation on the hydraulic head (Figure 10a). All wells show an increase in the water table owing to the recharge following the storms. This is more evident at the well located 20 km from Sisal (W4), which shows an increase of more than 2 m following the passage of Gamma and Delta, reaching a maximum level on October 7. It is worth noting that coastal wells W5 and W7a do not show such an increase in the water table due to confined aquifer conditions that do not allow rapid infiltration of the precipitation (Figure 10b); however, the storm effects can be seen in the change in amplitude (decrease) of the tide caused by the increase in aquifer discharge. The southern limit of such confinement is not well known, hence back-barrier flooding might*

*occur when the water table exceeds the confinement level south of the aquifer, preventing the hydraulic head in well W7a and W5 from increasing further (Perry, 1989; Pino 2011)."*

Forecast modelling. There is a mention of statistical fit, less bias and significantly high correlation. But the p-value is not presented to support these claims. The correlation (C) is the same correlation coefficient (r)? If so, wouldn't it be more appropriate to use this denomination?

RESPONSE: Thank you for your observations. The manuscript has been revised, replacing "correlation" by "correlation coefficient". Regarding the statistical parameters, a new Table was included in the manuscript to illustrate our results (Table1 below and Table 4 in the revised ms).

Table 1. Statistical parameters at the 5% significance level. df: degree of freedom; STD: standard deviation; ci: confidence interval.

| Variables | t-test | df | STD | p-value | CI | |
|---|---|---|---|---|---|---|
| Temperature | 13.17 | 4 | 0.08 | 0.0001 | 0.35 | 0.54 |
| Wind Speed | 6.29 | 4 | 0.20 | 0.003 | 0.32 | 0.82 |
| Wind Direction | 2.96 | 4 | 0.25 | 0.042 | 0.02 | 0.64 |
| Atmospheric Pressure | 6.14 | 4 | 0.22 | 0.004 | 0.33 | 0.88 |
| Significant Wave Height | 6.01 | 4 | 0.24 | 0.004 | 0.34 | 0.93 |
| Wave direction | 4.99 | 4 | 0.22 | 0.008 | 0.22 | 0.75 |
| Sea Level | 3.12 | 4 | 0.24 | 0.036 | 0.04 | 0.64 |

Discussion

The discussion is excessively short and does not cover essential aspects of the manuscript.

RESPONSE: We thank the reviewer for pointing out this issue. All the reviewer's comments are considered for the revised version of the manuscript. See responses below.

For example, what would be the main limitations of the approach used? particularly numerical modeling?

RESPONSE: The numerical modelling presents limitations in forcing and incorporating terrestrial and atmospheric processes. The spatial and temporal resolution for the wind field associated with the storm passage is not high enough to capture the hydrodynamic response near the coast for such events that move/travel offshore. On the other hand, field observations suggest that the flooding on the barrier island's lee side was associated with the high precipitation and the increase in the water table. These two aspects were not incorporated into the numerical models. A new subsection has been included in the revised manuscript (Section 5.2 Limitations).

What effect does the error associated with topographic measurements (i.e. bathymetry, beach profiles, DTM) have on the overall results of the study?

RESPONSE: The uncertainty in the topography and bathymetry is important for the model implementation since it controls wave transformation and coastal flooding. However, we found that our DGPS measurement errors are significantly smaller than the observed changes in the beach and hence do not affect the conclusions reached in the present study. See Lines 545-546.

If the geographic location of the population with respect to the path of the hurricanes seems to be a crucial factor in the adverse effects, as indicated at the end by including a comparison with another place, why was this approach not included in the experimental design?

RESPONSE: We agree that the geographical location can play an important role in the observed impact. In the present work, we focus on obtaining high-resolution measurements that can help to calibrate numerical models that can be implemented in other areas. However, we believe that comparing field observations with other places along the northern Yucatan coast warrants future studies.

It was also repeatedly mentioned that anthropic structures marked the transition in morphological changes; then, could it be said that if the structures did not exist, the changes would be more moderate?

RESPONSE: We believe the impact of the structures is quite important in the study area as confirmed by aerial images and in situ observations over the past few years. Observed morphological changes are significantly smaller at beach profiles located away from the structures. Therefore, it is likely that, as the reviewer points out, if the structures did not exist, extreme changes would be more moderate.

Would these morphological changes be permanent (long-term) or temporary?

RESPONSE: We agree that this is an important aspect that can be addressed with our beach profile measurements. To investigate this, we analyzed the beach morphology changes occurring during the following year after the storm passage (Figure 6 below and Figure 15 in the revised ms). It was found that some morphological features remained, such as the increase in the beach elevation in some areas, whereas other changes smoothed out during the winter storm season. Moreover, the beach evolution presented significant alongshore differences including a net volume gain (P16), fast recovery (P7), and a slow recovery (P20) after the storm passage (see Figure 5b).

[Figure]

**Figure 6: (a) Volume change evolution for each beach profile indicating the storm period for Gamma and Delta (dashed lines) and (b) time series of volume change for selected transects with different post-storm behavior (P7-stable, P16-accretion, P-20-erosion/recovery).**

Therefore, the discussion must be expanded and restructured, so that the most relevant aspects of the manuscript are addressed.

RESPONSE: Following the reviewer's comments, the Discussion section has been re-structured in the revised manuscript to comment on coastal resilience (Section 5.1) and the limitations of the present work (Section 5.2). Furthermore, the Conclusions section has been re-written accordingly.

REFERENCES:

Bhatia, K. T., Vecchi, G. A., Knutson, T. R., Murakami, H., Kossin, J., Dixon, K. W., & Whitlock, C. E. (2019). Recent increases in tropical cyclone intensification rates. Nature Communications, 10(1), 1–9. https://doi.org/10.1038/s41467-019-08471-z

Emanuel, K. (2017). Will global warming make hurricane forecasting more difficult? Bulletin of the American Meteorological Society, 98(3), 495–501. https://doi.org/10.1175/BAMS-D-16-0134.1

Emanuel, K. (2021). Atlantic tropical cyclones downscaled from climate reanalyses show increasing activity over past 150 years. Nature Communications, (1), 1–8. https://doi.org/10.1038/s41467-021-27364-8

Kossin, J. P., Emanuel, K. A., Vecchi, G. A. (2014). The poleward migration of the location of tropical cyclone maximum intensity. Nature, 509(7500), 349–352. https://doi.org/10.1038/nature13278

Kossin, J. P., Emanuel, K. A., Vecchi, G. A. (2014). The poleward migration of the location of tropical cyclone maximum intensity. Nature, 509(7500), 349–352. https://doi.org/10.1038/nature13278

Response to Anonymous Referee #2 comments:
Hazard assessment and hydrodynamic, morphodynamic, and hydrological response to Hurricanes
Gamma and Delta, on the northern Yucatan peninsula (MS No.: nhess-2022-113)
Alec Torres-Freyermuth et al.

The manuscript describes the results of a study on the effects and impacts of windstorms
Delta and Gamma, on a barrier island near the Yucatan coast through analyses of
environmental, morphological, anthropogenic, and ecological observations.

Overall the manuscript presents an impressive amount of data analyses. It provides fresh
insight into the very critical aspects of parameters are collectively influenced by the storms
and play significant roles in how barrier islands respond to extreme coastal events.

RESPONSE: We thank the reviewer for his/her comments.

My comments are minor:

1. English language here and there needs grammatical corrections. Please run the entire
   manuscript through copy editing and make sure there are no errors.

RESPONSE: The manuscript has been thoroughly revised by a native English speaker and
a professional editorial software (Grammarly) to ensure no errors in the revised version.

2. All symbols need to be defined, for instance, parameters noted in Line 132 are not
   defined.

RESPONSE: Thank you for pointing out this issue. We have ensured that all symbols are
defined in the revised version.

3. Line 134, why the low band is chosen to be 1/48? Why not 1/24?

RESPONSE: To ensure that the tidal signal is eliminated from the ADCP measurements,
we decided to cut the data at all frequencies higher than 48 hours. Cutting at 24 hours
leaves some information from the tidal signal based on our experience. See Lines 178-182.

4. Some tables are distorted such as Table 3.

RESPONSE: The format of all tables has been revised and homogenized.

Response to Ali Farhadzadeh comments:
Hazard assessment and hydrodynamic, morphodynamic, and hydrological response to Hurricanes Gamma and Delta, on the northern Yucatan peninsula (MS No.: nhess-2022-113)
Alec Torres-Freyermuth et al.

The manuscript describes the results of a study on the effects and impacts of windstorms Delta and Gamma, on a barrier island near the Yucatan coast through analyses of environmental, morphological, anthropogenic, and ecological observations.

Overall the manuscript presents an impressive amount of data analyses. It provides fresh insight into the very critical aspects of parameters are collectively influenced by the storms and play significant roles in how barrier islands respond to extreme coastal events.

RESPONSE: We thank Ali for his comments.

My comments are minor:

1. English language here and there needs grammatical corrections. Please run the entire manuscript through copy editing and make sure there are no errors.

RESPONSE: The manuscript has been thoroughly revised by a native English speaker and a professional editorial software (Grammarly) to ensure no errors in the revised version.

2. All symbols need to be defined, for instance, parameters noted in Line 132 are not defined.

RESPONSE: Thank you for pointing out this issue. We have ensured that all symbols are defined in the revised version.

3. Line 134, why the low band is chosen to be 1/48? Why not 1/24?

RESPONSE: To ensure that the tidal signal is eliminated from the ADCP measurements, we decided to cut the data at all frequencies higher than 48 hours. Cutting at 24 hours leaves some information from the tidal signal based on our experience. See Lines 178-182.

4. Some tables are distorted such as Table 3.

RESPONSE: The format of all tables has been revised and homogenized.